# Confinement-enhanced valorization of contaminants in electrified hydrogenation membranes for water purification

Yuyang Kang [1,2], Zhenao Gu [1,2,3] ✉, Wei Zhang [1,2], Baiwen Ma[1,2], Chenghai Lu[1,2], Chengzhi Hu [1,2] & Jiuhui Qu [1,2]

Electrocatalytic hydrogenation offers an environmentally benign approach for contaminant valorization, but suffers from sluggish mass and electron transfer. Electrified membranes (EMs) represent an effective strategy to address these challenges, yet their structure-performance relationship remains inadequately understood. Here, we develop EMs featuring atomically dispersed Ru sites, enabling the efficient hydrogenation of nitrate, trichloroacetic acid, and phenol. A volcano-shaped relationship is observed between electrocatalytic activity and pore size. The EM with a pore size of 7 μm (EM$_7$) achieves 94% nitrate removal within 55 s, exhibiting over 97% selectivity towards ammonium and a 2.5 times higher kinetic constant (2.7 min$^{-1}$) than that of EM with 80 μm pores (EM$_{80}$). However, further reducing the pore diameter to 4 μm diminishes performance. Multiphysics simulations reveal that smaller pores enhance mass transfer but worsen current distribution uniformity. Elucidating this spatial confinement effect offers a guiding design principle of cost-effective electrodes for sustainable wastewater treatment.

Oxidized and chlorinated compounds such as nitrate (NO$_3^-$), phenols, and haloacetic acid are commonly found in contaminated surface water, groundwater, and industrial effluents. Even at trace concentrations, they pose a significant threat to aquatic ecosystems and human health[1,2]. For instance, a high concentration of nitrate can lead to the eutrophication of water bodies[3,4], while excessive NO$_3^-$ in drinking water would induce adverse health effects such as methemoglobinemia[5]. It is estimated that more than 100,000 tons of nitrate compounds are discharged into water annually in the United States[6]. On the other hand, NO$_3^-$ serves as a raw material for the synthesis of ammonia through electrocatalytic hydrogenation (ECH) reaction[7,8], with the product being a valuable chemical used as fertilizer, chemical precursor, and green hydrogen carrier[9–11]. Similarly, the hydrogenation of toxic phenol could generate environmentally benign cyclohexanone and cyclohexanol[12], which are important

industrial feedstocks for the manufacturing of nylon[13]. In this regard, ECH of contaminants has aroused great interest as pollution control and resource recovery can be simultaneously realized.

The electrode-electrolyte interface plays a crucial role in the electrocatalytic processes, as mass diffusion, charge transfer, and chemical conversion are integrated within this region[14,15]. Mass diffusion is a pivotal step in electrocatalytic reactions, especially when treating low concentrations of contaminants[16]. Even with vigorous agitation, the hydrodynamic boundary layer of common plate electrodes is thicker than 100 μm, which significantly impedes the diffusion of reactants[17,18]. Besides, considering that most of the oxidized contaminants are negatively charged, the electrostatic repulsion at the cathode-electrolyte interface poses another substantial challenge to the diffusion and adsorption of reactants[14]. To address such issues, spatial confinement has emerged as an effective strategy[19,20]. Over the past

[1]Key Laboratory of Environmental Aquatic Chemistry, State Key Laboratory of Regional Environment and Sustainability, Research Center for Eco-Environmental Sciences, Chinese Academy of Sciences, Beijing, China. [2]University of Chinese Academy of Sciences, Beijing, China. [3]National Engineering Research Center of Industrial Wastewater Detoxication and Resource Recovery, Beijing, China. ✉e-mail: zagu@rcees.ac.cn

decade, electrified membranes (EMs) with diverse functionalities have been developed, wherein reactions are spatially confined within multi-scale pore networks[21,22]. EMs enhance reaction selectivity and reduce energy consumption[21], fostering their widespread application in fields such as chemical synthesis[23,24] and redox flow batteries[25]. Moreover, by compressing the diffusion boundary layer, EMs substantially improve the mass transfer rate of trace contaminants, offering considerable potential for water treatment and contaminant valorization[26,27].

Although EMs have proved their effectiveness in the removal and valorization of water contaminants, the design principle remains obscure due to a lack of mechanistic understanding on the spatially confined processes[28,29]. For instance, the impact of pore size, one of the crucial structural parameters, on the electrocatalytic performance is still unclear (Fig. 1a)[30]. The reported EMs vary in terms of catalysts and pore geometry, making it almost impossible to quantitatively assess the structure-performance relationships. Besides, the spatial distribution of reactants and interfacial current within the pores is largely unknown, preventing effective optimization of microstructure for the interfacial coupling of mass diffusion and electron transfer[31]. This lack of understanding on the spatial confinement effect hampers the development of highly efficient EMs and electrochemical devices.

Here, we developed EMs integrating ECH and membrane filtration to achieve efficient pollutant remediation and resource recovery from wastewater. Atomically dispersed Ru species were utilized as the electrocatalyst, contributing to fast electron transfer for the reductive hydrogenation of various contaminants, including nitrate, tri-chloroacetic acid (TCAA), and phenol. Remarkably, a volcano-shaped relationship between ECH activity and membrane pore size was observed in the range of 4 to 80 μm. An elaborately designed multi-physics model combining mass transfer, current distribution, and electron transfer was constructed to reveal the ECH mechanism in the spatially confined micropores. A compromise between accelerated mass transfer and decreased reactive depth was evidenced when reducing the pore size, thus contributing to an optimal structure. This study provides valuable insights into the fundamental understanding on the structure-performance relationships of EMs for sustainable water treatment.

## Results

### Characterization of EMs with atomically dispersed Ru

The EMs were constructed by employing porous Ti filters as the substrates, where reactions would be confined in the interconnected microchannels (Fig. 1b and Supplementary Fig. 1). TiO$_{2-x}$ nanosheets were densely fabricated on the side walls of the microchannels (Fig. 1c), upon which Ru catalyst was loaded. These nanosheets exhibited a lateral dimension of around 500 nm and a thickness lower than 10 nm (Supplementary Fig. 2), significantly increasing the specific surface

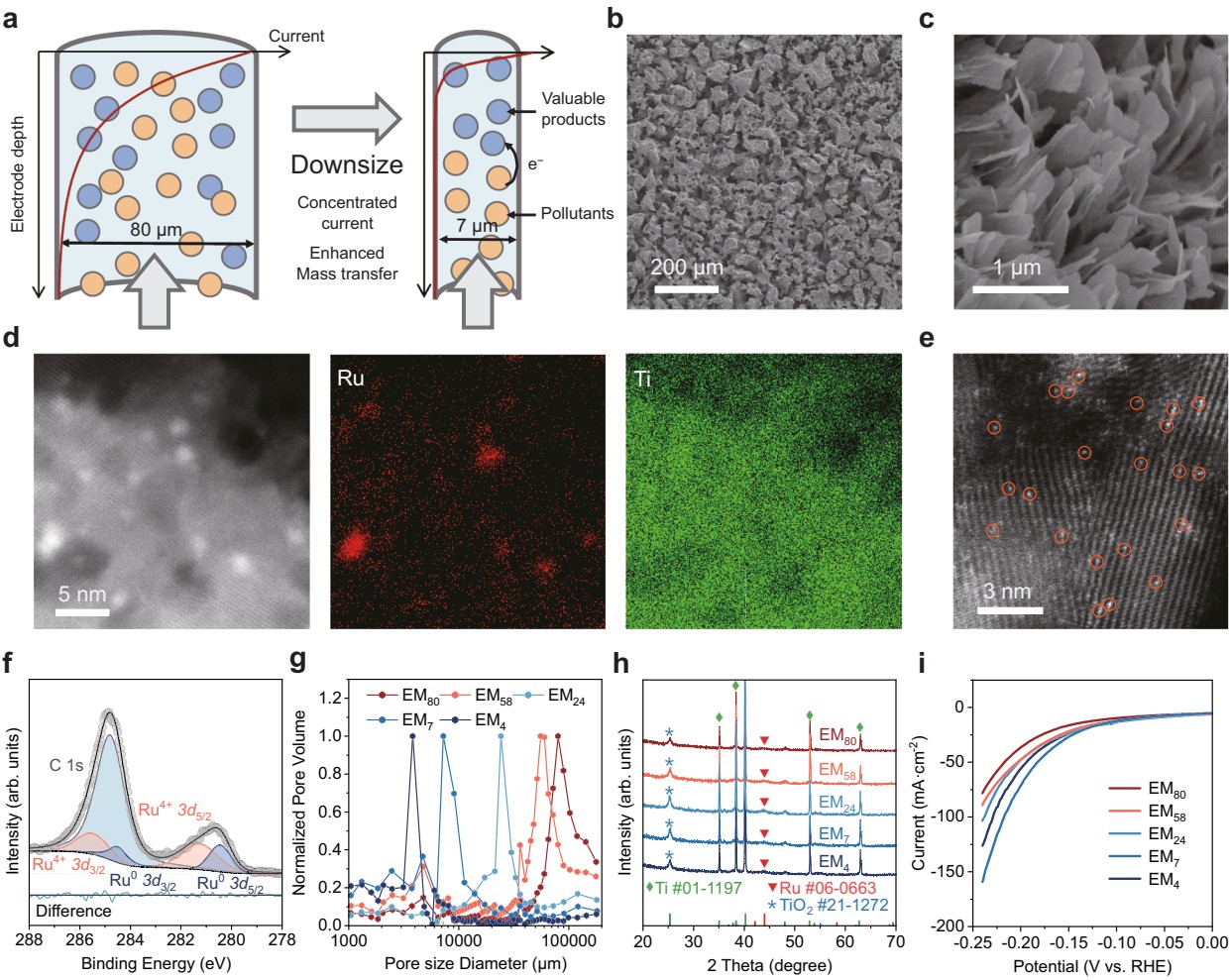

**Fig. 1 | Schematic and characterization of electrified membranes (EMs).**
**a** Schematic of the spatial confinement effect on EMs. **b** Scanning electron microscopy (SEM) image of the macroscopic pores in EM$_4$. **c** SEM image of the densely fabricated nanosheets on the EMs. **d** HAADF-STEM image of Ru/TiO$_{2-x}$ nanocatalyst and the corresponding elemental mapping images. **e** HAADF-STEM image of the single-atom (SA) Ru species on TiO$_{2-x}$ nanosheets, where some of the SAs are highlighted by orange circles. **f** High-resolution Ru 3$d$ XPS spectrum of the EMs. **g** Pore sizes of the EMs determined by mercury intrusion method. **h** XRD patterns of the EMs. **i** LSV curves of the EMs with different pore sizes. Electrolyte: 0.1 M Na$_2$SO$_4$.

area. Well-resolved fringe spacings of 0.35 nm can be observed on the high-resolution transmission electron microscopy (HRTEM) image (Supplementary Fig. 3), which are indexed to the (101) plane of anatase $TiO_2$. Notably, $TiO_{2-x}$ nanosheets contained numerous oxygen vacancies, as confirmed by electron spin resonance spectroscopy (ESR) and X-ray photoelectron spectroscopy (XPS) measurements (Supplementary Figs. 4 and 5). These defects can increase the electron conductivity and stabilize the highly dispersed Ru nanocatalysts. As evidenced by the high-angle annular dark-field scanning transmission electron microscopy (HAADF-STEM) images, a mixture of Ru nanoclusters and single atoms was uniformly loaded on $TiO_{2-x}$ nanosheets (Fig. 1d, e). The average size of the Ru nanoclusters was about 1.5 nm (Supplementary Fig. 3). Combined with the dense loading of single-atom Ru species, it can be verified that a high dispersion of Ru metal was achieved on the defective $TiO_{2-x}$ nanosheets. In addition, several Ru single atoms were adjacent to Ru clusters (Supplementary Fig. 3), which may an exhibit electronic synergy effect and improve the electrocatalytic performance[32]. The Ru $3d$ XPS spectrum reveals comparable amounts of $Ru^{4+}$ and $Ru^0$ species (Fig. 1f)[12,33,34]. The presence of $Ru^{4+}$ mainly originates from the coordination between the Ru single atoms and the adjacent O atoms[35].

The $Ru/TiO_{2-x}$ EMs with different pore sizes were fabricated. As determined by the mercury intrusion method, all EMs exhibited a standard monomodal pore size distribution, with modal pore diameters ranging from 4 to 80 μm (Fig. 1g). According to the log-normal distribution fitting results (Supplementary Fig. 6)[36], the standard deviations of the pore sizes for each EM were $3.8 \pm 0.4$, $7.3 \pm 1.4$, $24.3 \pm 1.6$, $58.6 \pm 10.5$, and $80.5 \pm 14.6$ μm, respectively. Accordingly, the EMs are denoted as $EM_4$, $EM_7$, $EM_{24}$, $EM_{58}$, and $EM_{80}$, respectively. The relatively uniform pores provided massive microchannels for accelerated mass and electron transfer. The X-ray diffraction (XRD) analysis revealed that the EM exhibits a weak diffraction peak at 44.0° (Fig. 1h), corresponding to the (101) planes of metallic Ru (JCPDS#06-0663). The broad half-peak width further underscored its ultra-small size and high dispersion. Meanwhile, strong XRD peaks at 25.3° and 52.8° were observed, which correspond to anatase $TiO_2$ (JCPDS#21-1272) and metallic Ti substrate (JCPDS#01-1197), respectively. It is noteworthy that the crystal structure displayed little variation across different pore sizes.

Subsequently, a series of electrochemical measurements were conducted to evaluate the properties of the EMs. The electrochemical double-layer capacitance ($C_{dl}$), directly related to the electrochemically active surface area (ECSA), was determined via cyclic voltammetry (Supplementary Fig. 7). A comparison of $C_{dl}$ values revealed that the ECSA increased as the pore size decreased and $EM_4$ exhibited the largest effective surface area, which is consistent with the Brunauer–Emmett–Teller (BET) analysis (Supplementary Fig. 8 and Supplementary Table 1). All the EMs demonstrated extremely low series resistance values, below 2 Ω (Supplementary Fig. 9), confirming the high conductivity of the defective $TiO_{2-x}$ nanosheets[37]. To further assess their electrocatalytic activity, linear sweep voltammetry (LSV) was performed (Fig. 1i). While the current densities generally increased with surface area, they did not scale proportionally. For instance, although $EM_4$ exhibited an ECSA 2.8 times that of $EM_{80}$ (Supplementary Table 1), its cathodic current density at $-0.2$ $V_{RHE}$ (56 mA cm$^{-2}$) was only 51% higher than that of $EM_{80}$ (37 mA cm$^{-2}$). Notably, $EM_7$ exhibited a current even higher than that of $EM_4$, despite possessing a smaller ECSA. This discrepancy suggests the existence of electrochemical nonuniformity within the micropores at relatively high current densities, where the full ECSA cannot be efficiently utilized[31].

## Nitrate reduction performance on EMs with varying pore sizes

Using a single-pass reaction system (Fig. 2a), we evaluated the nitrate reduction performance of EMs with varying pore sizes under an identical current density of 39.3 mA cm$^{-2}$ based on the geometrical area. A feed solution containing 1 mM nitrate was used to simulate the treated wastewaters[21]. The operating potentials of the EMs were nearly identical, falling within a narrow range of $-0.22$ to $-0.25$ $V_{RHE}$ (Supplementary Table 2). As the pore size decreased, $EM_7$ exhibited the highest catalytic efficiency, achieving a 94% removal of nitrate within a nominal hydraulic residence time ($HRT_{geom}$) of 55 s (Fig. 2b). However, only 66% of the nitrate was reduced on $EM_{80}$. The corresponding pseudo-first-order kinetic constant reached 2.7 min$^{-1}$ for $EM_7$ (Fig. 2c), which is 2.5 times that of $EM_{80}$ (1.1 min$^{-1}$). Polishing the catalytic layers on the surfaces showed no notable impact on their activity, and $EM_7$ continued to outperform $EM_{80}$ (Supplementary Fig. 10). This result indicates that the superior performance of the small-pore EM is primarily attributed to accelerated intrapore mass transfer driven by shorter diffusion distances. Notably, further reduction in pore size resulted in a decline in catalytic performance, evidenced by the relatively lower kinetic constant on $EM_4$ (2.1 min$^{-1}$) in comparison to $EM_7$. Considering that the local Ru loading density on $EM_4$ was lower than that on $EM_7$ due to difference in surface area, we fabricated another $EM_4$ with doubled Ru loading (from 0.91 to 1.82 mg cm$^{-2}$). No significant change in catalytic performance was observed (Supplementary Fig. 11), thereby ruling out the local Ru loading density as a determining factor.

Thereafter, the performance of the EMs in reducing nitrate at varying concentrations was assessed. The nitrate reduction rate increased markedly as its concentration decreased (Supplementary Fig. 12), primarily due to the enlarged ratio of active sites to nitrate molecules. At a nitrate concentration of 0.1 mM, the kinetic constant for $EM_7$ (6.0 min$^{-1}$) was four times that of $EM_{80}$ (1.5 min$^{-1}$). However, at a higher nitrate concentration of 10 mM, the kinetic constant of $EM_7$ was only 1.6 times that of $EM_{80}$. These results indicate that the advantage of $EM_7$ is more prominent at lower concentrations, where mass transfer limitations are more significant. Moreover, $EM_7$ consistently exhibited the highest reaction kinetics across all tested concentrations, further confirming its superiority over the other EMs. To further elucidate the mass transport capabilities of the EMs, $Cu^{2+}$ was employed as an electrochemical probe, as its reduction current effectively reflects mass transfer toward the catalyst surface[38,39]. The results showed that the observed mass transport rate ($R_{obs}$) increased significantly as the pore size decreased (Supplementary Fig. 13). Notably, $EM_4$ exhibited the highest mass transport performance among all EMs, reaching the convection limit (i.e., $Cu^{2+}$ is fully consumed at a specific flux) across the entire tested range. This paradox, where enhanced mass transfer coincides with diminished catalytic activity, implies that electron transfer processes may also play a critical role in determining the overall performance of the EMs.

The electron transfer mechanisms on the EMs were subsequently investigated. Generally, the electrochemical reduction of nitrate proceeds via either direct electron transfer (DET) or indirect pathway mediated by atomic hydrogen (H*)[12]. We used tert-butyl alcohol (TBA), a quencher of H*, to quantify the contribution of indirect electron transfer. Upon the introduction of TBA, there was no significant decrease in the nitrate removal kinetics of the EMs (Fig. 2d and Supplementary Fig. 14). The indirect electron transfer mediated by H* only accounted for 6% to 9% of the overall kinetics. Although the EMs exhibited substantial capacities for H* generation (Supplementary Fig. 15), nitrate reduction may primarily proceed via a proton-coupled electron transfer mechanism without significant involvement of H* species[40]. LSV tests under varying nitrate concentrations showed a marked increase in reduction current with increasing nitrate levels (Supplementary Fig. 16a). Notably, $EM_7$ exhibited significantly higher reduction currents than many previously reported electrocatalysts, even at relatively lower nitrate concentrations (Supplementary Fig. 16b and Table 3), highlighting its superior electrocatalytic activity. These results suggest that DET is the predominant reduction pathway, with no significant variation observed among the different EMs. From an

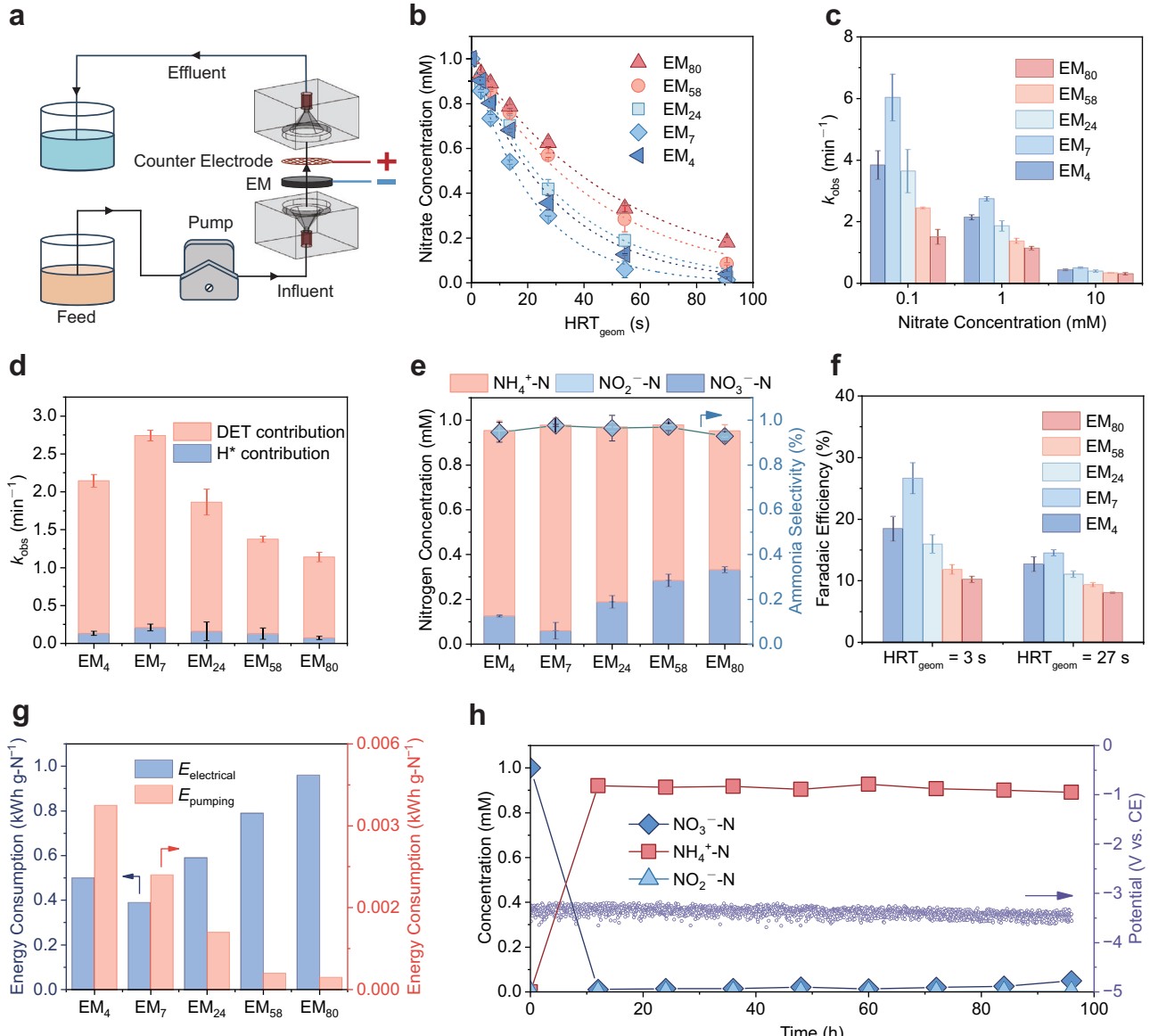

**Fig. 2 | Nitrate reduction performance on EMs. a** The single-pass filtration assembly used to conduct all the electrochemical experiments. **b** Removal of nitrate on EMs with different pore diameters. The dotted lines represent the pseudo-first-order kinetic regression curves ($R^2 > 0.99$). **c** Pseudo-first-order kinetic constants of nitrate reduction on the EMs at concentrations of 0.1, 1, and 10 mM. **d** Contributions of direct electron transfer (DET) and indirect electron transfer mediated by H* to the reaction kinetics. **e** Distribution of nitrogen species in the permeate (left axis) and $NH_4^+$ selectivity (right axis) on the EMs at a hydraulic residence time ($HRT_{geom}$) of 55 s. **f** FEs for nitrate reduction on the EMs at $HRT_{geom}$ values of 3 and 27 s. Nitrate concentration: 1 mM. **g** Energy consumption for nitrate reduction on the EMs. $E_{electrical}$ and $E_{pumping}$ represent the electrical and pumping energy, respectively. **h** Long-term nitrate reduction stability test of $EM_7$ with 1 mM nitrate at an $HRT_{geom}$ of 91 s. Current density: 39.3 mA cm$^{-2}$, reaction conditions: 0.1 M Na$_2$SO$_4$, pH = 7. Error bars represent the standard deviation from at least two independent tests.

application perspective, the high ratio of DET on the EMs would mitigate the negative influence of coexisting substances, which typically consume H* and compete for nitrate reduction[41].

The nitrate reduction products on the EMs within an $HRT_{geom}$ of 55 s were also analyzed. Ammonium ($NH_4^+$) was identified as the main product in the effluent, while little nitrite was detected for all the EMs (Fig. 2e). Remarkably, a high $NH_4^+$ selectivity of 97% was achieved on $EM_7$, indicating the high selectivity of Ru sites. Meanwhile, a relatively lower $NH_4^+$ selectivity of 92% was reached on $EM_{80}$. Based on the product analysis, the faradaic efficiencies (FE) for nitrate reduction were calculated. At an $HRT_{geom}$ of 3 s (Fig. 2f), the FE reached 28% on $EM_7$, indicating a higher electron utilization efficiency. In comparison, the FEs on $EM_4$ and $EM_{80}$ were 18% and 10%, respectively. Prolonging the $HRT_{geom}$ would reduce the FEs, mainly due to the

overconsumption of nitrate and decreased flux. Nonetheless, the superiority of $EM_7$ remains evident, exhibiting a nearly doubled FE (15%) at 27 s $HRT_{geom}$ compared to $EM_{80}$ (8%). When the nitrate concentration was increased to 10 mM, all EMs showed enhanced FEs, with $EM_7$ reaching a peak value of 80% under a 7 s $HRT_{geom}$ (Supplementary Fig. 17). As mass transfer limitations are alleviated at higher concentrations, the discrepancies in FEs among EMs become less pronounced.

The energy consumption of the EMs required to achieve 80% nitrate removal efficiency was calculated (Fig. 2g and Supplementary Table 4). Notably, $EM_7$ and $EM_{80}$ required $HRT_{geom}$ of 35 s and 85 s, respectively, to reach this target. Under these conditions, the transmembrane pressure on $EM_7$ reached 6.0 kPa (Supplementary Fig. 18), which is 8.5 times higher than that of $EM_{80}$ (0.7 kPa). However, the

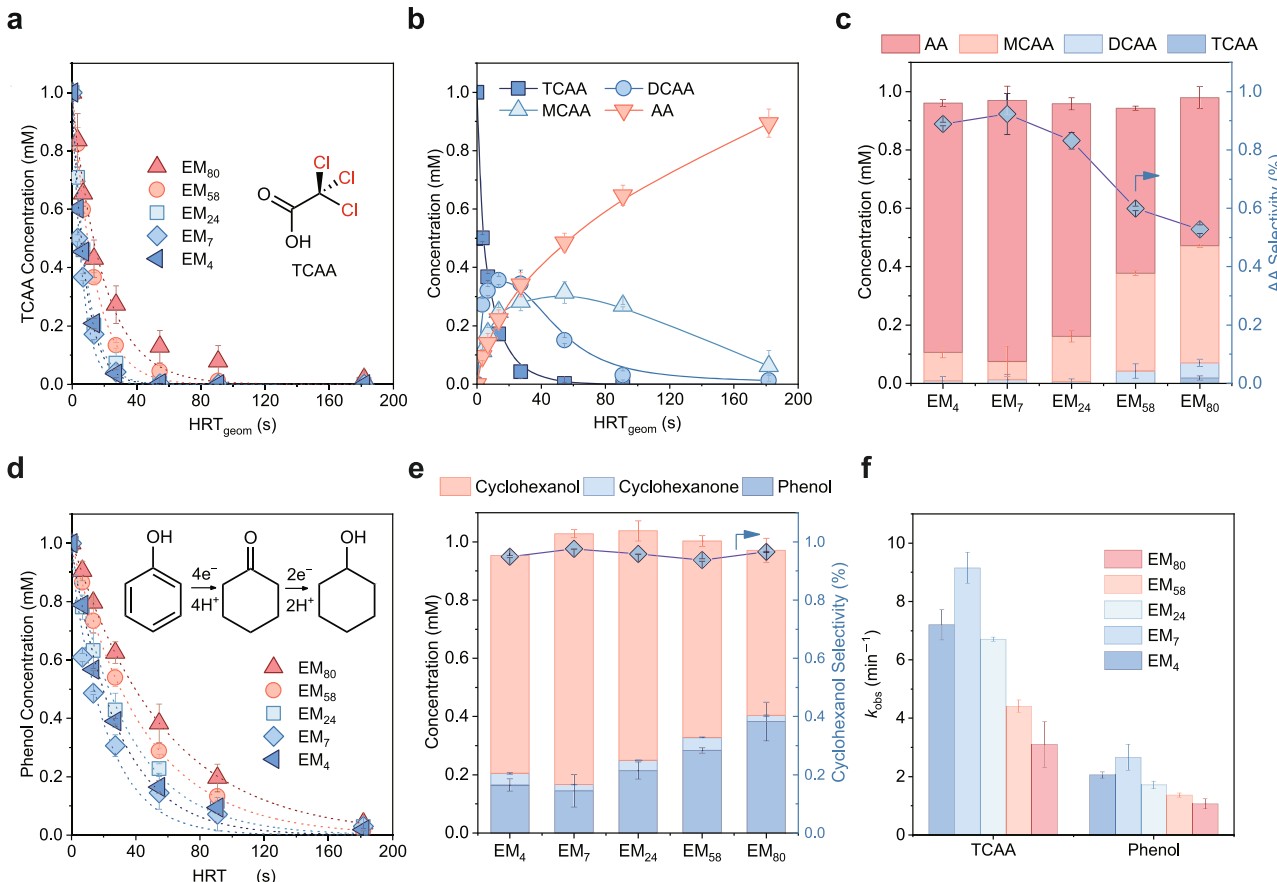

**Fig. 3 | TCAA dichlorination and phenol hydrogenation performance on EMs.**
**a** Removal of TCAA on the EMs. The dotted lines represent the pseudo-first-order kinetic regression curves ($R^2 > 0.98$). **b** Concentration evolution of TCAA and its dichlorination products on $EM_7$. The solid lines depict the schematic variation in concentration. **c** Distribution of TCAA reduction products in the permeate (left axis) and AA selectivity (right axis) on the EMs at an $HRT_{geom}$ of 55 s. **d** Removal of phenol on the EMs. The dotted lines represent the pseudo-first-order kinetic regression curves ($R^2 > 0.96$). **e** Distribution of phenol reduction products in the permeate (left axis) and cyclohexanol selectivity (right axis) on the EMs at an $HRT_{geom}$ of 55 s. **f** Pseudo-first-order kinetic constants of TCAA dichlorination and phenol hydrogenation on the EMs. The dashed lines in panels **a** and **d** represent the pseudo-first-order kinetic regression curves. Current density: 39.3 mA cm$^{-2}$, reaction conditions: 0.1 M Na$_2$SO$_4$, pH = 7. Error bars represent the standard deviation from at least two independent tests.

contribution of pumping energy per unit of nitrate removed is negligible compared to electrical energy. For instance, the pumping energy for $EM_7$ was 0.003 kWh g-N$^{-1}$, accounting for only 0.7% of the total energy consumption (0.39 kWh g-N$^{-1}$). $EM_7$ exhibited the lowest total energy consumption among the EMs due to its fast reaction kinetics. The above results indicate that there is an optimal pore size for the EM at around 7 μm. Further reduction in pore size not only leads to a lower kinetic constant but also increases the pumping energy requirements. The long-term durability of $EM_7$ was then investigated by a chronopotentiometry test at a fixed $HRT_{geom}$ (91 s). During a 96-h continuous operation, the catalytic performance of $EM_7$ remained rather stable (Fig. 2h). The Ru concentration in the effluent consistently remained below the detection limit (0.1 ppb) of inductively coupled plasma mass spectrometry (ICP-MS), indicating negligible Ru leaching. The conversion efficiency of nitrate remained consistently high, exceeding 95%, with NH$_4^+$ selectivity surpassing 92%. Additionally, almost no nitrite was detected in the effluent. A stable applied potential of 3.4 V was also observed, further underscoring the robust stability of $EM_7$ for nitrate reduction.

## Versatile applications of EMs for contaminant valorization

To explore the versatility of EMs in the conversion and valorization of common organic contaminants, we conducted additional ECH experiments using 1 mM TCAA and phenol as model pollutants, respectively. Under negative potentials, the hydrodechlorination

reactions of TCAA occurred, thus sequentially producing dichloroacetic acid (DCAA), monochloroacetic acid (MCAA), and acetic acid (AA) (Supplementary Fig. 19). As displayed in Fig. 3a, more than 50% TCAA was removed within 3 s on $EM_7$, and over 95% TCAA removal was achieved at an $HRT_{geom}$ of 27 s. In comparison, less than 17% of TCAA was eliminated on $EM_{80}$ within 3 s. The corresponding kinetic constant for $EM_7$ reached 9.2 min$^{-1}$, much higher than that of $EM_{80}$ (3.1 min$^{-1}$) (Fig. 3f). Interestingly, further reducing the EM pore size to 4 μm led to a decrease in TCAA hydrodechlorination kinetics, exhibiting a similar trend to that observed with nitrate (Fig. 2c). Such a volcano-type dependence of activity on pore size indicates the significant role of the confinement effect in regulating the electrochemical reactions. Subsequently, the reduction products of TCAA on $EM_7$ were analyzed. With an increase in $HRT_{geom}$, the TCAA concentration decreased rapidly, while the concentration of DCAA and MCAA increased and peaked at 0.36 mM and 0.31 mM, respectively (Fig. 3b). Further increasing the $HRT_{geom}$ led to a decline in DCAA and MCAA concentrations, indicating their roles as intermediates in subsequent hydrodechlorination reactions. More than 92% of TCAA was converted into AA as the final product at an $HRT_{geom}$ of 182 s, corresponding to a total dechlorination efficiency of 94% (Fig. 3c). In contrast, the dechlorination efficiency on $EM_{80}$ was less than 80% over the same duration, with both MCAA and AA as the main products (Supplementary Fig. 19). As a result, $EM_7$ consistently achieved significantly higher FEs than the other EMs (Supplementary Fig. 20). These results indicate that

that while spatial confinement can improve mass transfer, excessively small pore size can negatively impact pollutant removal.

Thereafter, the efficacy of EMs in the hydrogenation of toxic phenol was assessed. The hydrogenation products were usually cyclohexanone and cyclohexanol, which are important chemicals with significant industrial values and low toxicity[42]. As shown in Fig. 3d, f, the removal of phenol also follows a volcano-type dependence on the pore size of EMs, mirroring the trends observed with nitrate and TCAA. More than 69% of phenol was converted on $EM_7$ at an $HRT_{geom}$ of 27 s, which is higher than those on $EM_4$ (61%) and $EM_{80}$ (37%). Consequently, $EM_7$ achieved the highest kinetic constant (2.7 min$^{-1}$) among the EMs for phenol removal. The relatively slow reaction kinetics of phenol, compared to nitrate and TCAA, may be attributed to its larger molecular size, which hinders adsorption on the atomic Ru sites[43,44]. The product distributions among the EM systems were also analyzed. Phenol first undergoes a 4-electron reduction to cyclohexanone (intermediate), followed by a 2-electron reduction to cyclohexanol. At an $HRT_{geom}$ of 55 s, the concentration of cyclohexanol, the deeply hydrogenated product, reached 0.86 mM on $EM_7$ (Fig. 3e and Supplementary Fig. 21). Meanwhile, the concentration of cyclohexanone stood at only 0.02 mM, demonstrating the high hydrogenation activity of such an intermediate on the active sites. A long-term phenol hydrogenation experiment was subsequently conducted. $EM_7$ exhibited over 88% phenol removal and 94% selectivity towards cyclohexanol during the entire 96-h operation period, further validating its superior catalytic activity and stability (Supplementary Fig. 22). Based on these results, flow-through electrocatalytic reduction on EMs is proven to be a promising technology for contaminant detoxification and valorization. More importantly, the confinement effect emerges as a critical factor impacting the electrocatalytic efficiency.

## Confinement effect on electron transfer of EMs

To gain an insight into the confinement effect on EMs, multiphysics simulations were performed to investigate the mass transfer of nitrate and electrochemical processes within membrane pores (Fig. 4a). Five serpentine channels with diameters of 4, 7, 24, 58, and 80 μm were modeled to simulate the EMs, which are denoted as $Sim_4$, $Sim_7$, $Sim_{24}$, $Sim_{58}$, and $Sim_{80}$, respectively. An electrolyte solution containing 1 mM nitrate was introduced into the channels, under laminar flow conditions (Supplementary Table 5), with the influent stream flowing from the far ($x = 3$ mm) to the near ($x = 0$ mm) end. All the primary physiochemical processes were incorporated into this model, including ionic conductance, liquid flow, nitrate diffusion, interfacial electrochemical nitrate reduction, and hydrogen evolution reaction. To align with the experimental setup, both the inner channel surface and the exterior surface were designated as catalytic interfaces. The simulated current distribution along the microchannels is shown in Fig. 4b. Due to the fact that smaller channels possess larger surface areas, their current densities are relatively lower at a constant total current. It was observed that the channel walls near the counter electrode exhibited orders of magnitude higher current density than those at the far end. For instance, the current density was 17.3 mA cm$^{-2}$ at the nearest end of $Sim_{80}$, which is 2.8 orders of magnitude higher than that at the farthest end (0.03 mA cm$^{-2}$). This nonuniformity trend is also visually evidenced by an OH$^-$ fluorescence probe experiment at the Pt electrode interface (Supplementary Fig. 23 and Movie 1). The formation of the fluorescent boundary at the rear side of the electrode occurred significantly more slowly than at the front, highlighting the spatial variation in local electrochemical activity. Notably, such nonuniformity of current distribution is more pronounced for channels with smaller pores (Fig. 4b). Taking $Sim_4$ as an example, the current at the nearest end (8.4 mA cm$^{-2}$) is over 3.9 orders of magnitude higher than that at the farthest end (0.001 mA cm$^{-2}$). By roughly defining the electroactive region as the area where the local current density exceeds 1% of the maximum value, we can observe that the electroactive depth of $Sim_{80}$

is 750 μm (Supplementary Fig. 24), which corresponds to 25% of the total channel length (3 mm). Notably, the electroactive depth of $Sim_4$ is considerably smaller (230 μm), suggesting that the majority of the electrochemical reactions take place near the counter electrode.

As pore size decreases, nitrate reduction current also becomes more nonuniform (Supplementary Fig. 25), which, together with the decreasing nitrate concentration along the channel, leads to the formation of a current peak. Reduction in pore size shifts this peak toward the near end due to enhanced mass transfer and increased current nonuniformity. The ratios of nitrate reduction current to the total current, or the theoretical current efficiencies, were also determined and illustrated in Fig. 4b. The current efficiencies at the far end are generally higher than those at the near end. This phenomenon can be attributed to the much lower current density and higher nitrate concentration at the far end[45,46]. The theoretical current efficiencies basically follow an increasing trend with the reduction of pore sizes, mainly due to the enhanced nitrate diffusion. The interfacial charges corresponding to nitrate reduction per unit volume of solution were then integrated along the flow direction, i.e., from $x = 3$ mm to $x = 0$ mm (Fig. 4c). More than 88% of the charge transfer occurs within the surface 1 mm layer, demonstrating the nonuniform reaction distribution in the microchannels. Notably, although the theoretical local current efficiency of $Sim_4$ is higher than other Sims, the cumulative charge in the deeper region (e.g., $x > 1$ mm) of $Sim_4$ is lower than them, owning to the more severe nonuniform current distribution. In the surface region, this value notably escalates and exceeds that of $Sim_{58}$ and $Sim_{80}$. Although $Sim_4$ exhibits higher current efficiency along its entire length, the excessive concentration of current at the nearest end ($x = 0$ mm) results in a lower total cumulative charge compared to $Sim_7$. The result consolidates that the adverse effect of confinement primarily stems from the current nonuniformity.

Based on the well-established transmission line model, the nonuniform current distribution is primarily attributed to the solution resistance ($R_s$), which is influenced by both the pore size and electrolyte conductivity (Fig. 4d)[47,48]. As the distance from the counter electrode increases, the cumulative $R_s$ grows, leading to a larger solution potential drop ($\Delta\varphi_s$). Although the electrode potential ($\varphi_m$) remains constant throughout the depth, this difference in $\Delta\varphi_s$ results in variations in the local overpotentials ($\eta$), calculated as $\eta = \varphi_m - \Delta\varphi_s - \varphi_e$, where $\varphi_e$ represents the equilibrium potential[49]. This spatial variation in overpotential ultimately causes the observed nonuniform current distribution within the EMs. Considering the critical role of electrolyte concentration on $R_s$, its impact on the current distribution was quantitatively simulated (Supplementary Fig. 26). It is observed that increasing the electrolyte concentration significantly reduces the ratio of current at the nearest end to that at the farthest end (Supplementary Fig. 27). For instance, the near-to-far current ratio of $Sim_{80}$ decreased from $6.4 \times 10^3$ to 42 when the electrolyte concentration was increased from 0.02 M to 0.5 M. Similarly, the current ratio of $Sim_4$ also underwent a drastic decrease from $3.8 \times 10^5$ to $4.0 \times 10^2$. As the electrolyte concentration increases, the charge transfer of nitrate occurs throughout the entire length of the channel (Supplementary Fig. 28), indicating an expansion of the reaction zone. This phenomenon can be attributed to the reduced $R_s$, which decreases the potential drop ($\Delta\varphi_s$) across the pores.

To visualize the current distribution in the EM pores, Cu electrodeposition experiments were conducted. As shown in Fig. 4e, smaller pore sizes led to a more surface-concentrated Cu distribution. Upon increasing the electrolyte concentration, the Cu deposition depth was significantly enhanced. For instance, the Cu deposition depth on $EM_7$ in 0.1 M $Na_2SO_4$ was only 0.6 mm, which is substantially lower than that of $EM_{80}$ (1.6 mm). However, when the electrolyte concentration was raised to 0.5 M, the deposition depth on $EM_7$ increased markedly to 2.3 mm. To further validate the influence of electrolyte concentration on current density distribution, an optical observation experiment was

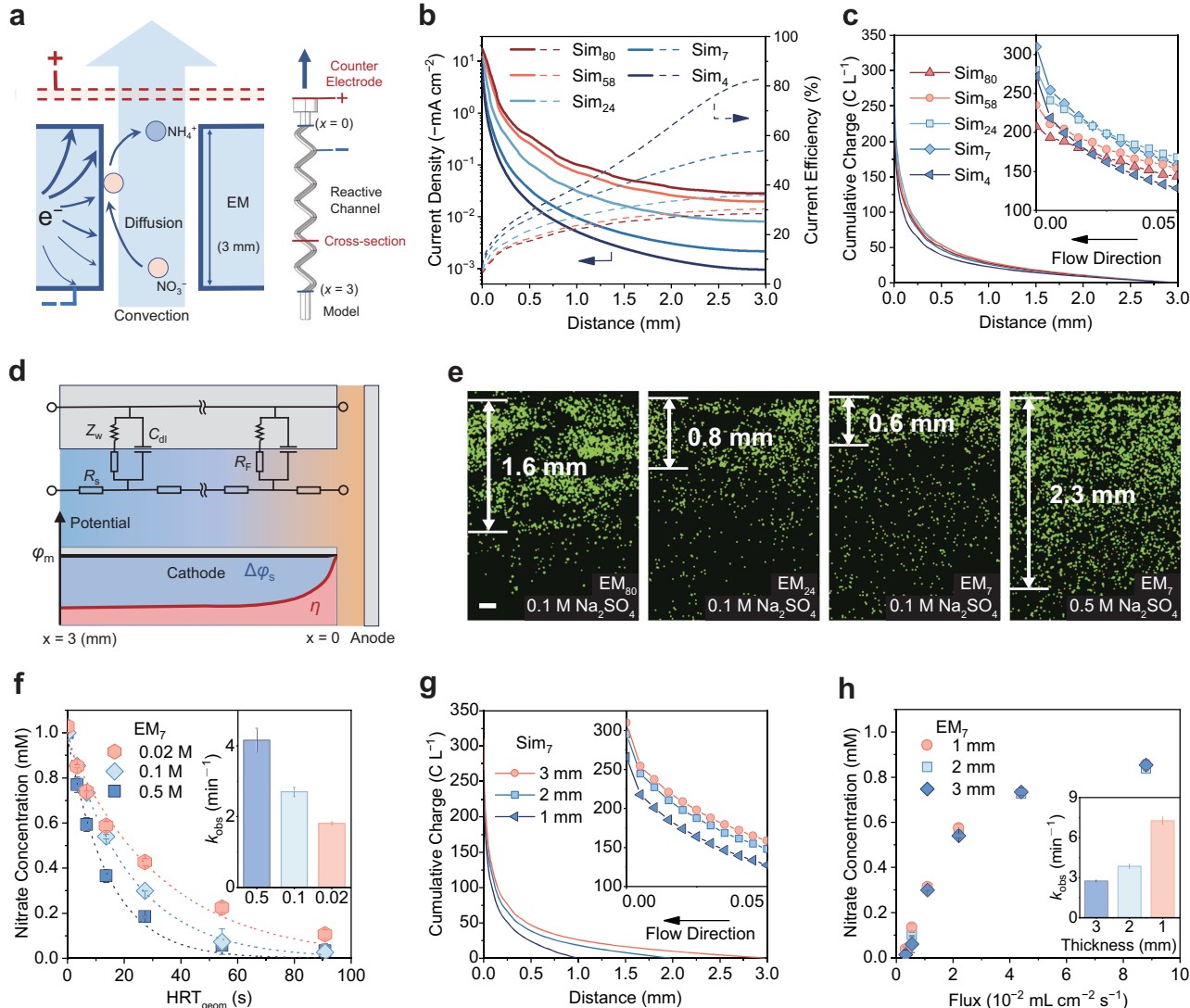

**Fig. 4 | Current distribution in the channel of EMs. a** Schematic of the simulation and the reaction process in the channel of EMs. **b** Interface current density distribution and the current efficiency of nitrate reduction on EM channels with different pore diameters at an $HRT_{geom}$ of 14 s. **c** Cumulative charge of nitrate reduction on EM channels with different diameters at an $HRT_{geom}$ of 14 s. **d** Equivalent circuit diagram and schematic illustration of potential distribution within EM pores, where $R_s$ represents the distributed resistance of the electrolyte, $R_F$ represents the Faradic charge transfer resistance, and $Z_W$ is the Warburg impedance. **e** Visualization of the current distribution on the EMs using the Cu elemental mapping (Scale bar: 0.2 mm). **f** Nitrate reduction performance on $EM_7$ at different concentrations of electrolyte. The dotted lines represent the pseudo-first-order kinetic regression curves ($R^2 > 0.97$). **g** Cumulative charge of nitrate reduction on $Sim_7$ with different lengths at an $HRT_{geom}$ of 14 s. **h** Nitrate removal performance on $EM_7$ with different thicknesses. The inset images in panels **c** and **g** show the enlarged cumulative charge distribution. The inset images in panels **f** and **h** show the corresponding pseudo-first-order kinetic constants. Current density: 39.3 mA cm-2. Error bars represent the standard deviation from at least two independent tests.

performed using a Pt electrode and a pH-sensitive probe to visualize the electrogenerated OH- ions (Supplementary Fig. 29 and Movie 2). At a high electrolyte concentration of 0.3 M, OH- ions were generated uniformly across the entire Pt electrode surface. In contrast, under low electrolyte concentration conditions (0.002 M), OH- production became highly localized in the region near the counter electrode. The above experimental results align well with the simulations, validating the credibility of this model. It is therefore speculated that the reactive region in EMs could be extended by increasing the electrolyte concentration. To verify this hypothesis, we further designed and conducted nitrate reduction experiments on $EM_7$ at different electrolyte concentrations. At a fixed current density (39.4 mA cm-2), when the electrolyte concentration was increased from 0.02 M to 0.5 M, the pseudo-first-order kinetic constant increased from 1.8 min-1 to 4.2 min-1 (Fig. 4f). However, increasing the electrolyte concentration does not significantly affect the catalytic performance in the flow-by

mode, where reactions are confined to the surface of the EM (Supplementary Fig. 30). This suggests that the main reason for the enhanced catalytic performance of EMs at high electrolyte concentrations is the change in $R_s$, which alters the current distribution within the pores. Therefore, in addition to mass transport, current distribution emerges as a critical factor governing the catalytic activity of EMs. Its optimization represents a promising strategy for enhancing reaction kinetics.

Based on the above analysis, electrochemical reactions primarily occur in the surface region, while the deep region is likely to be useless. To investigate this matter further, channels of different lengths (1 mm, 2 mm, and 3 mm) were simulated. Under an identical total current, the transferred charges corresponding to nitrate reduction were similar across channels of different lengths (Fig. 4g). Furthermore, $EM_7$ electrodes with thicknesses ranging from 1 to 3 mm were also fabricated and experimentally tested for nitrate reduction. It can be found that

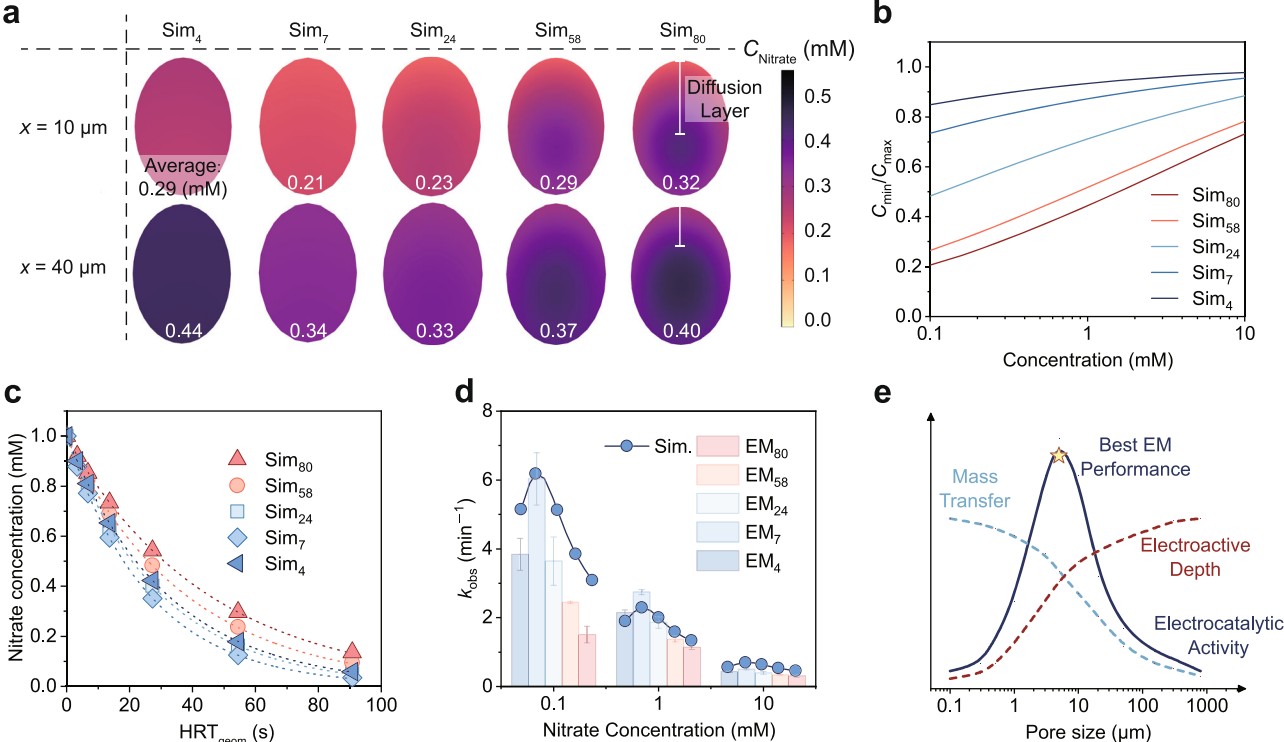

**Fig. 5 | Multiphysics simulation results of nitrate reduction on EMs. a** Visualized nitrate concentration in the cross-sections ($x = 10$ and $40\ \mu m$) of the EM channels at an $HRT_{geom}$ of 55 s. **b** Ratio of the lowest concentration ($C_{min}$) to the highest concentration ($C_{max}$) in the cross-section ($x = 10\ \mu m$) of the EM channels at an $HRT_{geom}$ of 55 s. **c** Removal of nitrate on EM channels with different pore diameters. The dotted lines represent the pseudo-first-order kinetic regression curves ($R^2 > 0.99$). **d** Comparison between the simulated and experimental pseudo-first-order kinetic constants of nitrate reduction. Sim. represents the simulated results. **e** Mechanistic illustration of the spatial confinement effect on EMs. Error bars represent the standard deviation from at least two independent tests.

the thickness of $EM_7$ had little impact on its catalytic performance (Fig. 4h and Supplementary Fig. 31). Even when the thickness (and consequently the Ru loading) was reduced to one-third, the 1mm-$EM_7$ still exhibited an extraordinary nitrate conversion rate. By comparing the $C_{dl}$ values of $EM_7$ with different thicknesses (Supplementary Fig. 32), it is evident that the ECSA of 1mm-$EM_7$ is significantly lower than that of 3mm-$EM_7$. The discrepancy indicates that the ECSA cannot be fully utilized in microchannels with relatively high current densities and low solution conductivities. These findings suggest that $EM_7$ of varying thicknesses share a similar electroactive thickness, high-lighting the potential to maintain high catalytic performance while reducing material usage and fabrication costs.

## Coupling of electron and mass transfer on EMs

The confinement effect of micropores in EM plays a crucial role in regulating the mass transfer and its coupling with electron transfer. To have an insight into these intricate processes, the steady-state concentration distribution of nitrate in the cross-section of simulated channels was calculated and visualized (Fig. 5a). Elliptic cross-section was created due to the serpentine channel structure, with the major axis reaching 1.4-fold the nominal diameter. Concentration polarization can be clearly observed in $Sim_{80}$, with the diffusion layer thickness estimated to be about $60\ \mu m$ in the near-surface region (e.g., $x = 10$ $\mu m$). Additionally, the polarization in the sub-surface region (e.g., $x = 40\ \mu m$) is less prominent, possibly due to the relatively lower local current density. With the reduction of pore sizes, the concentration polarization gradually diminishes. For instance, the nitrate concentration at the center of $Sim_4$ ($x = 10\ \mu m$) was $0.30\ mM$, almost identical to that near the channel wall ($0.28\ mM$). This implies that nitrate diffusion can be greatly accelerated by the confinement effect. It is also clearly shown that the nitrate reduction reaction primarily occurs near the pore opening. The smaller the pore size, the more

unevenly the reaction is distributed along the channel. By comparing the average concentrations at two cross-sections ($x = 40\ \mu m$ and $x = 10\ \mu m$), $Sim_4$ exhibited a concentration drop of $0.15\ mM$ (from $0.44\ mM$ to $0.29\ mM$), whereas $Sim_{80}$ showed a smaller decrease of $0.08\ mM$ (from $0.40\ mM$ to $0.32\ mM$). Notably, although $Sim_4$ exhibits a larger local concentration reduction, its effluent concentration remains higher than that of $Sim_7$ due to excessive surface-localized reactions (Supplementary Fig. 33). We then analyzed the concentration polarization of the EM channels at different inlet nitrate concentrations in the cross-section at $x = 10\ \mu m$ using the ratio of the minimum to maximum concentration ($C_{min}/C_{max}$) as an indicator (Fig. 5b). As the inlet concentration increased, the polarization within the channels significantly decreased, and the differences among Sims were reduced. $Sim_4$ and $Sim_7$ showed minimal concentration polarization at all concentrations, demonstrating enhanced mass transfer in small-pore EMs. Notably, at a concentration of $1\ mM$, reducing the pore size from $7\ \mu m$ to $4\ \mu m$ only slightly increased the $C_{min}/C_{max}$ value from $0.87$ to $0.93$, indicating that a pore size of less than $7\ \mu m$ is sufficient to meet mass transfer requirements (Supplementary Fig. 34). Therefore, current distribution plays a more dominant role within this pore size range.

Based on the simulated results, the outlet nitrate concentrations at different $HRT_{geom}$ values were obtained (Fig. 5c). The trend aligns well with the experimental data, with $Sim_7$ exhibiting the best nitrate removal performance. Due to the relatively shorter electroactive depth, nitrate reduction in $Sim_4$ is less efficient. Subsequently, pseudo-first-order kinetic constants were calculated based on the simulated effluent nitrate concentrations. The simulation results closely match the experimental values (Fig. 5d), confirming the accuracy of the model. $Sim_7$ exhibited the highest kinetic constant among the channels regardless of the influent nitrate concentration. For instance, the kinetic constants of $Sim_{24}$, $Sim_7$, and $Sim_4$ were $2.0\ min^{-1}$, $2.3\ min^{-1}$, and

1.9 min$^{-1}$ for 1 mM nitrate, respectively. With a decrease in influent nitrate concentration, all the kinetic constants increased. It is also noteworthy that the catalytic activity of the model's exterior surface has minimal influence on the overall performance (Supplementary Fig. 35), which is consistent with the experimental observations (Supplementary Fig. 10). This can be attributed to two key factors: (1) the exterior surface area is considerably smaller than the internal surface area within the pores, and (2) stagnant zones near the exterior surface substantially impede mass transfer, thereby limiting catalytic efficiency in these regions (Supplementary Fig. 36). Moreover, the simulated reaction kinetics under different thicknesses and electrolyte concentrations also align well with the experimental results (Supplementary Fig. 37). The high consistency between the simulated and experimental results indicates that this model is an instructive approach to unveil the microscopic reaction mechanism in the confined EM pores.

## Discussion

This study presents the design principle of EMs for efficient wastewater remediation and pollutant valorization by gaining mechanistic insights into the spatial confinement effect. EMs loaded with atomically dispersed Ru sites were fabricated, showcasing versatile activity for the hydrogenation of nitrate, TCAA, and phenol. Reducing the pore diameter of EM (e.g., $EM_7$ vs. $EM_{80}$) significantly decreases the diffusion lengths of reactants, leading to accelerated mass transfer and alleviated concentration polarization. However, the enhancement in mass transfer becomes less prominent once the pore size drops below a certain threshold (i.e., ~7 μm), as evidenced by both experimentally measured mass transfer rates (Supplementary Fig. 38) and simulated reaction kinetics at uniform current distribution (Supplementary Fig. 39). Meanwhile, the nonuniformity of current distribution along the channel depth was intensified as the pore size decreased, due to the increasing solution resistance inside the channel. As demonstrated by the batch mode experiments conducted under equivalent mass transfer conditions (Supplementary Fig. 40), such nonuniformity significantly reduces the catalytic performance of EMs with smaller pore sizes (e.g., $EM_4$ vs. $EM_7$). As a result of the compromise between mass transfer (favored in small-pore EM) and electron transfer (favored in large-pore EM), a volcano-shaped relationship between electrochemical activity and membrane pore size was observed (Fig. 5e). Consequently, there exists an optimal pore size for EMs, which is determined to be around 7 μm in our system.

Through combined simulations and experiments, we systematically analyzed the influence of EM architecture and electrolyte properties on catalytic performance. Although thickness directly impacts the surface area and catalytic loading, the benefit of increasing thickness is marginal, as it does not necessarily enhance reactive depth. Additionally, a high electrolyte concentration is essential for promoting current uniformity and expanding the effective reaction area. Porosity also plays a significant role in determining performance. When the porosity of $EM_7$ was reduced to 60% (matching that of $EM_4$) by selectively blocking a portion of its pores, a 15% decline in catalytic performance was observed (Supplementary Fig. 41), likely due to impaired current distribution and reduced reactive surface area. Nonetheless, $EM_7$ still outperformed $EM_4$ in reaction kinetics at the identical porosity, underscoring the critical contribution of spatial confinement effects. The impact of other key design parameters, including applied current, intrinsic catalytic activity, reactant diffusion coefficient, and electrode spacing, were also analyzed by simulation, respectively (Supplementary Fig. 42). As the applied current increases, faster mass transfer is required to balance electron transfer, which amplifies the advantage of smaller pores and may reduce the optimal pore size. Meanwhile, improvements in intrinsic catalytic activity enhance surface reaction rates but have minimal influence on the optimal pore size. Similarly, variations in diffusion coefficient and

electrode spacing show little effect on the optimal pore dimension. This in-depth understanding of the structure-performance relationship offers insights into the design of EMs with ultrahigh reaction kinetics, advancing beyond conventional catalyst development paradigms.

Collectively, our study reveals that while enhanced mass transport dominates the activity improvement at larger pore scales (e.g., $EM_7$ vs. $EM_{80}$), severe current nonuniformity becomes the primary limitation for smaller pores (e.g., $EM_4$). The electrochemical model provides a foundational framework that integrates current distribution, mass transfer, and electron transfer processes, enabling quantitative prediction of EM performance trends under varied conditions. For instance, the performance of emerging EMs (e.g., asymmetric EM) can be explored in silico prior to experimental investigation. Given the superior catalytic efficiency of EMs, this study presents an instructive approach for designing cost-effective EMs. These advancements have significant implications not only for wastewater treatment but also for resource recovery, fostering the development of more sustainable wastewater management frameworks.

To further refine and quantify the insights provided by this framework, future work should aim to decouple the quantitative contributions of mass transport and current distribution. From a computational perspective, more intuitive mathematical models could be developed to directly account for the interplay between pore structure, mass transfer, and current distribution, thereby facilitating the rational design of optimal pore architecture. To enhance the model's predictive capability, it is preferable to include the effects of bubbles, intrinsic reaction kinetics, and other relevant factors. Experimentally, an important research focus lies in the design of EM systems capable of isolating the individual influences of these interrelated processes. One promising approach is the fabrication of EMs with precisely tunable pore characteristics (e.g., size and porosity), using techniques such as laser etching or photolithography. On these well-defined platforms, tailoring the conductivity of electrolytes and electrode materials, combined with experiments performed under carefully controlled conditions (e.g., specific current densities and potentials), can help evaluate the influence of current distribution. Concurrently, in situ analytical methods, such as microfluidic chips, can be employed to quantitatively investigate the mass transfer at the electrode interface. Finally, extending studies to nanoscale pores (<100 nm) may also reveal distinct nanoconfinement effects (e.g., electric double layer overlap[50] and mesoscopic mass transfer[15]), which significantly influence both mass transport and local reaction kinetics, potentially reshaping pore-size optimization strategies.

## Methods
### Preparation of EMs

Ti substrates with diameters of 25 mm and thicknesses of 3 mm were made from Ti particles, which were pressed into shape and then sintered under vacuum. The pore size distribution is controlled by the size of the particle and the condition of the preparation process (e.g., pressure and sinter temperature). The Ti substrates underwent a cleaning procedure involving sequential immersion in acetone, ethanol, and deionized water for 30 min each. Subsequently, the cleaned substrate underwent etching in 10% oxalic acid at a temperature of 100 °C for 1 h. TiO$_2$ nanosheets were then synthesized by immersing the substrate in a 5 M NaOH solution at a temperature of 180 °C for 2 h within a Teflon-lined stainless steel autoclave. The resultant Na$_2$Ti$_3$O$_7$-coated Ti substrate was treated with 1 M HCl for 1 h to form the H$_2$Ti$_3$O$_7$ layer. Finally, the TiO$_2$ layer was produced through calcination at 400 °C for 2 h in air. The obtained electrode was submerged in a RuCl$_3$ solution and dried in a freeze-dryer (Scientz-10N/A, China). The final material was obtained after annealing in a hydrogen atmosphere at 300 °C for 2 h to reduce the Ru species and induce the formation of

oxygen vacancies in TiO₂ nanosheets (denoted as $TiO_{2-x}$). The loading mass density of Ru was $0.91\,mg\,cm^{-2}$ on each EM (unless otherwise mentioned). For comparison, we synthesized EMs with pore sizes of 4, 7, 24, 58, and 80 μm using the aforementioned method.

## Flow-through experiments

All electrochemical experiments were carried out employing a dead-end filtration setup, with membrane flux regulated through a peristaltic pump. A RuO₂/Ti mesh was used as the counter electrode. An Ag/AgCl electrode was used as the reference electrode. The solution flows sequentially from the bottom through the working electrode (EM) and then the counter electrode, which facilitates bubble removal. All electrochemical removal experiments were carried out using a CHI660E electrochemical workstation (CH Instruments) at a temperature of $30 \pm 1\,°C$. The reaction area, measuring $2.54\,cm^2$ was defined by a silicone gasket of 1 mm thickness. The electrolyte solution contained 1 mM nitrate (unless otherwise mentioned) and 0.1 M Na₂SO₄ (pH = 7). The flow-through experiments were conducted under a constant current ($39.3\,mA\,cm^{-2}$). The $HRT_{geom}$ was determined based on the geometric volume of the EMs ($0.76\,cm^3$) in order to correct for porosity variations among the EMs. The Cu electrodeposition experiment was performed in the potentiostatic mode, at $-0.85\,V_{Ag/AgCl}$ for 20 min, in a solution containing 50 mM Cu₂SO₄ and 0.1 M Na₂SO₄.

## Mass transfer determination

The $R_{obs}$ value on the electrode was determined using a well-established approach[38,39]. A solution containing 0.1 mM $Cu^{2+}$ and 0.1 M Na₂SO₄ was used as the electrolyte. The same assembly as the flow-through experiments was employed to conduct all the tests. The counter and reference electrodes were a Pt wire and Ag/AgCl electrode, respectively. All the electrochemical experiments were performed on a CHI660E electrochemical workstation at $30 \pm 1\,°C$. The reduction current of $Cu^{2+}$ was obtained at $0.26\,V_{RHE}$. The $R_{obs}$ value was calculated using Eq. (1)[51]:

$$R_{obs} = \frac{I}{zFAC_b} \tag{1}$$

where $I$ is the reduction current (A), $z$ represents the number of electrons transferred (2 for the reduction of $Cu^{2+}$), $F$ denotes the Faraday constant ($96,500\,C\,mol^{-1}$), $A$ is the geometry surface area of the electrode ($2.54 \times 10^{-4}\,m^2$), and $C_b$ is the bulk concentration of $Cu^{2+}$ ($0.1\,mol\,m^{-3}$).

## Characterization

The morphology of the prepared electrode was examined by a scanning electron microscope (Hitachi SU8000). ESR analysis was conducted using a Bruker A300 spectrometer. XPS spectra were acquired on a Thermo ESCALAB250Xi spectrometer with a monochromated Al Kα X-ray source. XRD measurements were conducted on a Bruker Diffractometer with Cu Kα radiation. The distribution of pore sizes was analyzed using the mercury intrusion method (AutoPore IV 9500). The porosity of reactive EMs was determined by Archimedes' Drainage Method (Supplementary Table 1). All the electrochemical measurements were carried out on a Gamry Interface 1000 electrochemical workstation. The morphologies and energy dispersive spectroscopic elemental mapping of Ru species were recorded on an aberration-corrected HAADF-STEM (JEM-ARM300F). The OH⁻ fluorescence probe experiment was conducted on a confocal laser scanning microscope (OLYMPUS IX83ZDC).

## Multiphysics simulation

The multiphysics analysis was performed using COMSOL Multiphysics 5.3a. EM channels with diameters of 4, 7, 24, 58, and 80 μm were modeled. The model length was set to be 3 mm, which is the same as

the thickness of EMs in the experiment. An inlet region and an outlet region were added to the channel. The range of $HRT_{geom}$ set in the EM channels (3–182 s) was generally consistent with the experiment (3–182 s). The concentration distribution of nitrate was solved by the "transport of diluted species" module. The mathematical model of the mass transfer includes Fick's law, the convection process, and the reaction process. Due to the variation of current density with depth within the EM, the "secondary current distribution" module was used to investigate the current distribution in the flow-through system. The Butler-Volmer equation and Ohm's law were used to solve for the current distribution and reaction process in the channels. The applied current was determined according to the porosity and current in the experiment (Supplementary Table 1). The detailed parameters are presented in Supplementary Table 6.

## Analysis methods

The kinetics of contaminant removal was fitted using the pseudo-first-order kinetics according to the following equation:

$$C_t = C_0 e^{-\frac{kt}{60}} \tag{2}$$

where $t$ is the $HRT_{geom}$ (s), $C_t$ represents the effluent concentration at a given $HRT_{geom}$, $C_0$ is the initial concentration, $k$ represents the pseudo-first-order kinetic constant ($min^{-1}$).

The concentration of $NO_3^-$, $NO_2^-$, and $NH_4^+$ were measured using UV-Vis spectroscopy (Hitachi U3900) as described in Supplementary Information. Potential Ru leaching was analyzed using ICP-MS (Agilent 7850). The concentration of phenol and its hydrogenation products were measured using high-performance liquid chromatography (Agilent 1260) and gas chromatography–mass spectrometry (Agilent 7890A/5975C), respectively. The concentration of TCAA and its reduction products were measured using an ultra-performance liquid chromatography system (Thermo Fisher TSQ Altis) equipped with a mass detector.

## Data availability

All data needed to evaluate the conclusions in the paper are present in the paper and/or the Supplementary Information. Source data are provided with this paper.

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

## Acknowledgements

This work was supported by the National Natural Science Foundation of China (52125003 and 52388101 to C.H., 52470098 to Z.G., 52322001 to B.M.), the International Partnership Program of the Chinese Academy of Sciences (032GJHZ2023019MI to C.H.), and Beijing Nova Program (20250484841 to Z.G.).

## Author contributions

Z.G. and C.H. came up with the original idea. Z.G., C.H., and J.Q. supervised the project. Y.K., Z.G., B.M., C.H., and J.Q. designed the experiments. Y.K. and Z.G. performed the experiments and the characterizations. Y.K., Z.G., and W.Z. analyzed the data. B.M. and C.L. discussed the results. Y.K., Z.G., C.H., and J.Q. wrote the paper. All authors discussed, commented on, and revised the manuscript.

## Competing interests

The authors declare no competing interests.
