## [Transparent Peer Review file · Nature Communications]

Confinement-enhanced valorization of contaminants in electrified hydrogenation membranes for water purification

Corresponding Author: Professor Zhenao Gu

Version 0:

Reviewer comments:

Reviewer #1

(Remarks to the Author)

The manuscript by Gu et al. demonstrates the valorization of various contaminants (nitrate, phenol, TCAA) in wastewater using electrified membranes (EMs) embedded with atomically dispersed Ru species. The authors systematically investigate the influence of electrode structure and operational conditions on treatment efficacy through a combination of experiments and modeling. The discovery of a volcano-shaped relationship between pore size, mass diffusion, and electron transfer processes is notable. The results are comprehensive and contribute to a deeper understanding of the reaction mechanisms within electrode micropores. The study has the potential for eventual publication after addressing the following issues:

1. The BET analysis indicates that EM4 has a higher surface area than other EMs with larger pores. Given the same Ru loading (Line 453), this may result in a lower local mass loading density on EM4, potentially affecting the single atom/cluster ratio and corresponding catalytic activity. The authors should discuss the impact of this phenomenon on catalytic performance.
2. While the authors analyze the catalytic performance of electrodes with different pore sizes, it is unclear if reactions occurring on the electrode surface (outside the pores) were considered, as these may also contribute to the overall electrochemical reactions. How was this aspect accounted for in the simulations?
3. The authors claim that active hydrogen contributes minimally in the EM systems (Fig. 2d). Is this due to the confinement effect or the low activity of the Ru catalyst in producing active hydrogen? Clarifying this distinction would strengthen the discussion.
4. While the manuscript analyzes the impact of pore size on current distribution, the influence of pore size on mass transfer is less quantitatively addressed. Given that both current distribution and mass transfer significantly affect surface reactions, a detailed discussion on the latter is recommended.
5. Numerical simulations strongly support the experimental findings, but some critical details are not given. For example, the fluid flow type (laminar or turbulent) in the microchannels is not specified, which is a crucial factor influencing mass transfer. Additionally, the rationale behind the modeling parameters in Table S3 should be clearly demonstrated.
6. The reduction of TCAA produces MCAA as an intermediate (Fig. 3b), which is reported to be more toxic than TCAA (Environ. Sci. Technol. 2000, 34 (13), 2675-2683). The capability of EMs to achieve complete dechlorination should be further investigated and discussed.
7. The stability of the electrodes is a critical issue. The authors should provide data on Ru leaching during electrolysis to validate the long-term performance and robustness of the electrodes.
8. The water flow directions depicted in Figs. 1a and 2a appear inconsistent. Revising these diagrams for clarity is recommended.
9. Supplementing Fig. 3d with the reaction equations for phenol reduction would enhance the clarity of the results.
10. The explanation of reaction distribution in the microchannels (Lines 371–377) is somewhat ambiguous. The authors should revise these sentences for improved clarity and precision.

Reviewer #2

(Remarks to the Author)

This study prepares a range of electrified membranes with varying pore sizes including atomically dispersed Ru to

demonstrate reduction of a variety of pollutants, including nitrate, trichloroacetic acid, and phenol. They obtain a semi-volcano-shaped relationship between the electrocatalytic activity and the pore size, where the smallest pore size tested shows a decrease (different trend from the rest). While this study does contribute new data to the literature on flow-through electrode pore sizes and conversion trends across different contaminants, it is not clear that it provides any new fundamental insight. A major concern is that the explanation for the trend (and the modeling conducted) assumes that current varies through the depth of the membrane, but this should not be the case for a uniformly conductive monolithic electrified membrane (i.e., electrons should transfer freely throughout the conductive solid, and the current at any given point in the membrane should be the same). There are also many other potential issues in the choice of experimental parameters measured that are used to draw conclusions, as detailed below. It is possible that the trend is due to optimization of transport rates in a reactive system, as previously reported. I believe that it is important for the authors to consider and address these points detailed below to see if their trends and explanations still hold after these corrections.

MAJOR COMMENTS

1. Lines 86-89, TiO₂ is described as being a conductive substrate, but it's insulating. The authors need to be clearer about the electrical conductivity of the substrate, which will have significant implications for their major hypothesis / finding that the current varies through the depth of the membrane substrate.

2. Especially since the pore size of the EMs is so important, more information about the data collected on determining pore size, including standard deviation or a pore size distribution, should be provided.

3. P. 5-6, the authors discuss the lack of correspondence between BET surface area and LSVs. However, they should quantify the electrochemically active surface area, which they should be able to do from their cyclic voltammetry. This could eliminate the lack of correspondence that they observe, since there is a correlated trend.

a. The authors argue that differences in surface area in the membranes with different pore sizes cannot account for the differences in conversion activity (specifically, the trends are similar, but not exactly proportional). However, they use BET surface area as the relevant parameter. This is a measure of physical gas adsorption sites, but the relevant surface area is related to electrochemical activity. Therefore, electrochemically active surface area should be used as the relevant parameter to confirm whether these changes account for the differences and performance for the membranes with different pore sizes.

4. The authors do not seem to report any potential reference to a standard reference electrode. This will make the results impossible to compare with other nitrate reduction results in existing literature, or to have a sense of how large the potential and overpotential are.

5. Line 188, the HRT of seven seconds is used for the energy consumption calculation, but 80% nitrate conversion is assumed, which seems to require an HRT around 55 seconds. The considered conditions do not seem to be consistent for the parameters used in these calculations.

6. Lines 279-293, it is not clear why current would change with the depth of the channels. If the membrane is conductive, and if the material composition and therefore conductivity is uniform throughout, then the potential at any given point in the membrane should be the same as the potential at any other point. It seems as though the authors are treating the thickness of the membrane as if it were an electrolyte, which is vanishingly conductive to electrons, rather than as an electrode. However, electrodes are typically highly electrically conductive, meaning that delocalized electrons move freely and rapidly throughout the conductive material (by definition, an electrode is the conductive interface that allows free electron transfer). Therefore, the entire concept of current changing with the depth of the membrane, and therefore the explanation about why differences are observed for membranes with different pore sizes, does not seem to make sense. This is very important here: the key novel scientific ideas in this manuscript seem to rest on this idea.

7. P. 14, the authors discussed increasing electrolyte concentration as a way to illustrate the effects of pore size on current and activity. However, it is obvious that increasing electrolyte concentration decreases solution resistance, and therefore increases current in the cell. This does not seem to have anything to do with membrane pore size.

8. Fig. 4h and related discussion: the authors present results of increasing the membrane thickness based on residence time. Since the residence time is calculated based on membrane thickness, the effect of the greater thickness is effectively canceled out, as the residence time has a normalizing effect. Therefore, this figure actually seems to show that the full length of the pores is utilized in the reaction. Otherwise, a constant residence time for different pore lengths would result in significant changes in activity, if the hypothesis of the authors is correct that only the area near the surface of the membrane is utilized for reactions. These results seem to disprove the proposed mechanisms by the authors, actually showing the opposite of the key scientific claim.

9. Lines 465-466, the HRT is determined based on the geometric volume. However, it should be determined by the pore volume, which corresponds to the void volume that is actually occupied by water flowing through the membrane. This will vary with pore size.

10. What are the flow rates of the permeate through the membranes? Since the HRT is kept constant and the pore size is changing, presumably the flow rate for each is also changed. Therefore, the obtained volcano relationship could actually be due to optimization of mass transfer, considering both diffusion and convection, as was previously shown for electrified

11. SI line 59, what concentration of nitrate is used? Is it the initial concentration, or the concentration of nitrate removed? For the energy consumption for conversion, it should be the latter. In the main text line 185, it seems like the nitrate conversion is assumed to be 80%, but the actual conversion should be included for calculating the energy consumption, not an assumed conversion.

12. It seems as though the reason for the FE for ammonia changing with pore size could be actually due to activity (FE for different products changes for different conversation rates). It would be more accurate to instead compare FE for nitrate conversion rather than for one product since the conversion changes.

13. The authors find an optimal pore size for their system, but it is not clear how this finding could be easily extrapolated to any other systems, as there are no quantitative relationships or fundamental/generalizable parameters drawn from their study that seemed like it could be easily extended to other systems.

MINOR COMMENTS

1. Some references do not seem to be appropriate for the contents, such as references 23, 24 and 25 in line 57.

2. In figure S 13, which product is the current efficiency referring to?

Version 1:

Reviewer comments:

Reviewer #1

(Remarks to the Author)

This reviewer is satisfied with the response, and has no further comment.

Reviewer #2

(Remarks to the Author)

The authors have done a good job of addressing most of the comments, specifically through running additional experiments and providing more details and explanations. Crucially, they conducted ECSA measurements, referencing electrode potentials to a reference electrode, and improved consistency for parameters used to calculate energy consumption. More importantly, I believe that they have now provided good evidence that the effect of pore size on current distribution through the electrified membranes influences reaction performance, which is the key finding and novelty in this work. I have some remaining comments, below, but I believe that if they are addressed (especially comment 1) then the manuscript would be suitable for publication in Nature Comms.

1. Reviewer 1, comment 4: The authors conclude that since mass transfer improves with decreasing pore size, and since EM4 with the smallest pore size shows reduced reaction activity, then other factors such as nonuniformity in current could explain the trends. However, the results show that EM4 reaches the convection limit. Therefore, could it be that EM4 is simply additionally mass transport limited by convection, leading to the lower reaction activity? Further, it appears that mass transport must account substantially for the reported decrease in activity with increasing pore size that occurs for EM7 – EM80. The authors should address how much this trend is due to changes in mass transport vs. current distribution (for example, by modifying parameters to ensure the same mass transport rate through all EMs; the same HRT in membranes that have different mass transport rates does not account for this).

2. The authors noted that they reduced TiO₂ in hydrogen at elevated temperature and that the sample becomes conductive. Therefore, I believe that the confusion arises because this is a titanium suboxide: for example, many EM papers generate Ti₄O₇ as a highly conductive Magneli phase titanium oxide material. It is therefore now inaccurate to refer to the synthesized material (after the hydrogen treatment) as TiO₂. They could refer to it instead as reduced titania/titanium oxide or titanium suboxide.

3. The authors have done a great job of adding more explanation and experimental verification of the variation in current through the depth of the EMs, which is central to the key findings and novelty. As a follow-up to this: the authors should consider that the changes in limiting diffusion current through the EMs likely varies with EM pore size (presumably the reactions are faster in the smaller pores, so the limiting diffusion current will likely increase as well), which should affect the current distribution. In other words, pore size influences mass transport, so the influence of mass transport on current density should also change with pore size.

4. Reviewer 2, comment 9: The authors explain that calculating HRT based on pore volume rather than geometric surface area would result in an unfair comparison, as smaller pores results in lower flux at the same HRT (although I believe they meant that it results in higher flux). Instead of using geometric volume, which does not in fact represent the volume of water passing through the membrane, perhaps the comparisons should be based on flow rate, which should be more consistent

among the membranes with different pore volumes. The authors also do not seem to explain why geometric volume is a valid way to calculate HRT, other than that it's an alternative that avoids concerns with using pore volume. Specifically, what is the physical meaning of HRT based on geometric volume in these porous systems?

5. Reviewer 2 comment 11: If HRT is used to determine volumetric flow rate, but the volume used to calculate HRT is the geometric volume and not the pore volume, then the obtained flow rate should be inaccurate, resulting in inaccurate calculation of the energy efficiency.

Version 2:

Reviewer comments:

Reviewer #2

(Remarks to the Author)

The revised manuscript addresses the points raised well. The additional experiments and simulations that separately evaluate the influences of mass transfer and current distribution now provide convincing evidence for the key result of optimal pore size that balances these effects. However, these tradeoffs are still not quantitatively evaluated in this manuscript, and it seems like mass transport is still the dominant aspect determining the pore size effect on conversion of reactants in solution. Therefore, I believe that the manuscript would be strengthened by adding some brief discussion about future work being required to quantitatively evaluate the respective contributions of these two phenomena in EMs. For example, there may be a more complex relationship among pore size effects, mass transport, and current distribution (which are all interrelated), and it may be that reducing pore size into the nm scale leads to different effects. Overall, though, this manuscript does provide convincing evidence for the concept of current nonuniformity being a limiting factor for smaller pore sizes leading to enhanced activity, which is a novel and valuable contribution for this field.

Point-by-point response to the reviewers' comments

Title: “*Confinement-enhanced valorization of contaminants in electrified hydrogenation membranes for water purification*”

Manuscript ID: NCOMMS-24-81216

We extend our sincere gratitude to the reviewers for their insightful comments and constructive suggestions, which have significantly contributed to enhancing the quality of this work. We have carefully and systematically responded to all the points raised. The reviewers' comments are in ***bold italic font*** and our revisions are in **blue font**. We have also highlighted the revised text in **blue** in the main text. Provided below are our detailed responses to each point.

REVIEWER COMMENTS

Reviewer #1 (Remarks to the Author):

The manuscript by Gu et al. demonstrates the valorization of various contaminants (nitrate, phenol, TCAA) in wastewater using electrified membranes (EMs) embedded with atomically dispersed Ru species. The authors systematically investigate the influence of electrode structure and operational conditions on treatment efficacy through a combination of experiments and modeling. The discovery of a volcano-shaped relationship between pore size, mass diffusion, and electron transfer processes is notable. The results are comprehensive and contribute to a deeper understanding of the reaction mechanisms within electrode micropores. The study has the potential for eventual publication after addressing the following issues:

We sincerely appreciate your recognition of this work and your constructive suggestions for improving the manuscript. Your insightful comments will greatly help us improve the quality of this paper. In response to your comments, we have made substantial revisions, primarily focusing on the following three key aspects:

- (1) The mass transfer performance of reactants within the EM channels was quantitatively evaluated via Cu^{2+} electro-reduction reaction. The results revealed a clear trend that EMs with smaller pores exhibit significantly

enhanced mass transfer compared to those with larger pores, and no expectation was observed. This finding further supports our conclusion that the volcano-shaped activity curve arises from current distribution rather than mass transfer limitations.

- (2) A more detailed explanation of the model's underlying assumptions, construction process, and parameter determination has been provided.
- (3) Additional validation experiments were conducted (e.g., investigating the influence of the single-atom/cluster ratio and reactions on the EM surface) to further substantiate the conclusions.

The inclusion of these data significantly strengthens the rigorousness and comprehensiveness of our study. A point-by-point response to your comments is as follows.

1. The BET analysis indicates that EM4 has a higher surface area than other EMs with larger pores. Given the same Ru loading (Line 453), this may result in a lower local mass loading density on EM4, potentially affecting the single atom/cluster ratio and corresponding catalytic activity. The authors should discuss the impact of this phenomenon on catalytic performance.

We appreciate your valuable comment. Since all EMs have the same Ru loading (0.91 mg cm^{-2}) but differ in BET surface area, the smaller-pore EMs exhibit lower local Ru loading densities. To investigate the potential impact of local Ru loading density on catalytic performance, we doubled the Ru loading on EM₄ to 1.82 mg cm^{-2} (from the original 0.91 mg cm^{-2}) and compared their catalytic activity and selectivity. The results showed no significant improvement in either catalytic efficiency or ammonia selectivity compared to the original loading. In other words, even when the local Ru loading of EM₄ became comparable to that of EM₇, its performance remained notably inferior.

We have added the relevant statements in the main text and Supplementary Information.

Line 159 – 163 in the revised manuscript:

“Considering that the local Ru loading density on EM₄ was lower than that on EM₇ due to difference in surface area, we fabricated another EM₄ with doubled Ru loading (from

0.91 to 1.82 mg cm⁻²). No significant change in catalytic performance was observed (Supplementary Fig. 11), thereby ruling out the local Ru loading density as a determining factor.”

Page 17 in the revised Supplementary Information:

Supplementary Fig. 11. The nitrate removal performance on EM₄ with different Ru loading. (a) Removal of nitrate on EM₄ with different Ru loading. (b) Distribution of nitrogen species in the permeate (left axis) and NH₄⁺ selectivity (right axis) at an HRT of 55 s. Electrolyte: 0.1 M Na₂SO₄.”

2. While the authors analyze the catalytic performance of electrodes with different pore sizes, it is unclear if reactions occurring on the electrode surface (outside the pores) were considered, as these may also contribute to the overall electrochemical reactions. How was this aspect accounted for in the simulations?

Thank you for your suggestion. Indeed, reactions can occur on both the upper surface (adjacent to the counter electrode) and lower surface (facing away from the counter electrode) of the EMs. To ensure consistency between simulation and experiment, both exterior (upper and lower) surfaces were assigned identical reactivity to the pore surface in the model.

To quantify the contribution of the exterior surfaces to the overall reactivity, the exterior surfaces of EM₇ and EM₈₀ were polished to remove the surface Ru catalyst. Given the inherently lower activity of Ti than Ru, the current in the pores may increase after polishing. The results revealed only a slight improvement in electrocatalytic activity on the EMs after polishing (Supplementary Fig. 10), suggesting that the exterior surfaces contribute minimally to the total reactivity. Moreover, additional simulations

incorporating “surface polishing” (where the exterior surface were rendered non-reactive) were conducted. The modeling results confirmed that surface polishing had negligible impact on catalytic activity (Supplementary Fig. 34). Collectively, both the experimental and computational results demonstrate the insignificant role of the exterior surfaces on the electrochemical reactions.

Furthermore, the underlying mechanism was also analyzed. On the one hand, the exterior surface area is much smaller than the interior area in the pores. On the other hand, stagnant flow zones exist near the exterior surface (Supplementary Fig. 35), which would greatly impair the mass transfer and catalytic performance. We have included the relevant discussion in the revised main text and Supplementary Information.

Line 154 – 157 in the revised manuscript:

“Polishing the catalytic layers on the surfaces showed no notable impact on their activity, and EM₇ continued to outperform EM₈₀ (Supplementary Fig. 10). This result indicates that the superior performance of the small-pore EM is primarily attributed to accelerated intrapore mass transfer driven by shorter diffusion distances.”

Line 309 – 311 in the revised manuscript:

“To align with the experimental setup, both the inner channel surface and the exterior surface were designated as catalytic interfaces.”

Line 452 – 458 in the revised manuscript:

“It is also noteworthy that the catalytic activity of the model’s exterior surface has minimal influence on the overall performance (Supplementary Fig. 34), which is consistent with the experimental observations (Supplementary Fig. 10). This can be attributed to two key factors: (1) the exterior surface area is considerably smaller than the internal surface area within the pores, and (2) stagnant zones near the exterior surface substantially impede mass transfer, thereby limiting catalytic efficiency in these regions (Supplementary Fig. 35).”

Page 16 in the revised Supplementary Information:

Supplementary Fig. 10. The nitrate removal performance on (a) EM₇ and (b) EM₈₀ with surface polishing. Electrolyte: 0.1 M Na₂SO₄.”

Page 40 in the revised Supplementary Information:

Supplementary Fig. 34. Simulated nitrate removal performance on (a) Sim₇ and (b) Sim₈₀ with surface polishing. Polishing refers to setting the exterior surface of the model to be non-catalytic. Electrolyte: 0.1 M Na₂SO₄.”

Page 41 in the revised Supplementary Information:

Supplementary Fig. 35. Schematic of the flow field on the upper surface of EMs.”

3. The authors claim that active hydrogen contributes minimally in the EM systems (Fig. 2d). Is this due to the confinement effect or the low activity of the Ru catalyst in producing active hydrogen? Clarifying this distinction would strengthen the discussion.

Thank you for your comment. We employed electron spin resonance (ESR) spectroscopy to assess atomic hydrogen (H^*) generation capacity of EMs with varying pore sizes. The results revealed slightly stronger ESR signals from smaller pores compared to larger ones, aligning with their catalytic performance trends. This observation suggests that enhanced mass transfer facilitates the binding of 5,5-dimethyl-1-pyrroline N-oxide (DMPO, the trapping agent) and H^* .

However, based on the quenching experiment (Fig. 2d), the contribution of H^* appears to be minimally influenced by pore confinement effects. While H^* can act as an active species in nitrate reduction, the electrochemical reduction of nitrate primarily follows a proton-coupled electron transfer (PCET) mechanism (*Small Methods* **2020**, 4, 2000672; *Chem. Soc. Rev.* **2022**, 51, 2710–2758), bypassing direct H^* involvement.

In the LSV tests of EM₇ conducted under varying nitrate concentrations (Supplementary Fig. 16a), the reduction current increased significantly with higher nitrate levels, confirming the substantial contribution from direct electron transfer. Collectively, these results indicate that the limited contribution of H^* is intrinsically linked to the nitrate reduction mechanism on EMs.

We have added the relevant statements in the main text and Supplementary Information.

Line 187 – 195 in the revised manuscript:

“Although the EMs exhibited substantial capacities for H* generation (Supplementary Fig. 15), nitrate reduction may primarily proceed via a proton- coupled electron transfer mechanism without significant involvement of H* species⁴⁰. LSV tests under varying nitrate concentrations showed a marked increase in reduction current with increasing nitrate levels (Supplementary Fig. 16a). Notably, EM₇ exhibited significantly higher reduction currents than many previously reported electrocatalysts, even at relatively lower nitrate concentrations (Supplementary Fig. 16b and Table 3), highlighting its superior electrocatalytic activity. These results suggest that DET is the predominant reduction pathway, with no significant variation observed among the different EMs.”

Page 21 in the revised Supplementary Information:

Supplementary Fig. 15. DMPO spin trapping ESR spectra on EMs. Electrolyte: 0.1 M Na₂SO₄.”

Page 22 in the revised Supplementary Information:

Supplementary Fig. 16. (a) Linear sweep voltammetry (LSV) curves of EM₇ in the presence of different concentrations of nitrate.”

4. While the manuscript analyzes the impact of pore size on current distribution, the influence of pore size on mass transfer is less quantitatively addressed. Given that both current distribution and mass transfer significantly affect surface reactions, a detailed discussion on the latter is recommended.

Thank you for raising this important point. Our initial study lacked quantitative analysis of mass transport behavior across EMs with varying pore sizes, which is essential for interpreting the observed volcano-shaped activity curve. As also highlighted by **Reviewer 2 (Comment #10)**, this curve might arise from an optimization of mass transfer. To address this, we employed the methodology described in “*ACS Nano* **2019**, 13, 6998–7009” and “*Electrochim. Acta* **1990**, 35, 1369–1376”, employing Cu²⁺ as a probe. By pumping Cu²⁺ solutions through EMs at a fixed potential and varying flow rates, the mass transport rates were quantified by analyzing Cu²⁺ reduction currents (Supplementary Fig. 13).

The results revealed a clear increase in mass transport rates with decreasing pore sizes. At relatively low flow rates, Cu²⁺ was fully reduced across all EMs regardless of pore size, with the observed mass transport rates (R_{obs}) approaching the convection limit. However, at higher flow rates, larger-pore EMs exhibited mass transport rates far below the convection limit. Notably, EM₄ demonstrated superior mass transport rates at higher flow velocities, outperforming EM₇.

Furthermore, by transposing the Supplementary Fig. 13a, we obtained a curve consistent with the schematic mass transfer curve in Fig. 5e. At a fixed flow rate, decreasing pore size causes the mass transfer rate to gradually plateau (Supplementary

Fig. 37), at which point current distribution becomes dominant, as reflected in the volcano-shaped curve. In other words, even though the EM₄ exhibits the highest mass transfer rate, its catalytic performance is inferior to that of EM₇. Our findings demonstrated that this contradiction is primarily attributed to the current nonuniformity caused by spatial confinement.

We have added the relevant statements in the main text and Supplementary Information.

Line 172 – 180 in the revised manuscript:

“To further elucidate the mass transport capabilities of the EMs, Cu²⁺ was employed as an electrochemical probe, as its reduction current effectively reflects mass transfer toward the catalyst surface^{38,39}. The results showed that the observed mass transport rate (R_{obs}) increased significantly as the pore size decreased (Supplementary Fig. 13). Notably, EM₄ exhibited the highest mass transport performance among all EMs, reaching the convection limit across the entire tested range. This paradox, where enhanced mass transfer coincides with diminished catalytic activity, implies that electron transfer processes may also play a critical role in determining the overall performance of the EMs.”

Line 545 – 556 in the revised manuscript:

“Mass transfer determination

The observed mass transfer rate (R_{obs}) value to the electrode was determined using a well-established approach^{38,39}. A solution containing 0.1 mM Cu²⁺ and 0.1 M Na₂SO₄ was used as the electrolyte. The same assembly as the flow-through experiments was employed to conduct all the tests. The counter and reference electrodes were a Pt wire and Ag/AgCl electrode, respectively. All the electrochemical experiments were performed on a CHI660E electrochemical workstation at 30 ± 1 °C. The reduction current of Cu²⁺ was obtained at 0.26 V_{RHE}. The R_{obs} value was calculated using equation (1)⁵⁰:

$$R_{\text{obs}} = \frac{I}{zFAC_{\text{b}}} \quad (1)$$

where I is the reduction current (A), z represents the number of electrons transferred (2 for the reduction of Cu²⁺), F denotes the Faraday constant (96,500 C mol⁻¹), A is the geometry surface area of the electrode (2.54×10^{-4} m²), and C_{b} is the bulk concentration

of Cu^{2+} (0.1 mol m^{-3}).”

Page 19 in the revised Supplementary Information:

Supplementary Fig. 13. Mass transfer rate on EMs. (a) Observed mass transfer rate (R_{obs}) on EMs. (b) Comparison of R_{obs} between EM₄ and EM₈₀. Highlighted areas correspond to the contribution of surface area and pore size. Since EM₄'s ECSA is 2.8 times that of EM₈₀, the orange region represents the maximum contribution (2.8×) from the specific surface area to mass transport, while the blue region reflects the minimal contribution from pore structure. Experiments were conducted at 0.26 V_{RHE} in an electrolyte containing 0.1mM Cu^{2+} , and 0.1M Na_2SO_4 .”

Page 43 in the revised Supplementary Information:

Supplementary Fig. 37. Mass transfer rate on EMs at different fluxes as a function of HRT.”

5. Numerical simulations strongly support the experimental findings, but some critical details are not given. For example, the fluid flow type (laminar or turbulent) in the microchannels is not specified, which is a crucial factor influencing mass transfer. Additionally, the rationale behind the modeling parameters in Table S3 should be clearly demonstrated.

We sincerely appreciate the reviewer’s constructive suggestions. We have thoroughly discussed three key aspects of the model: its underlying assumptions, construction methodology, and parameter determination. For instance, the flow type within microchannels was defined as laminar, based on Reynolds number calculations ($Re < 1$ for all EMs, even at the highest flow velocity). Details have also been provided in the Supplementary Table 5. Additionally, a discussion on the rationale for parameter selection has been supplemented in the discussion section and Supplementary Table 6.

The basic assumptions and construction process have been supplemented as follows:

Page 3 – 4 in the revised Supplementary Information:

“The multiphysics analysis was simulated using COMSOL Multiphysics 5.3a. The pore diameter was simplified to the modal pore size observed in EM experiments, ignoring the actual pore size distribution. Accordingly, EM channels with diameters of 3.8, 7.0, 24.0, 58.0, and 80.0 μm were modeled, respectively. Outlet and inlet were connected to both ends of the channel, respectively. The distance between the inlet and outlet is 3 mm, consistent with the thickness of EM. All channel models were assumed to have the same tortuosity (tortuosity = 1.4²) and were simplified as regular cylinders. The channel was modeled with a certain curvature to simulate the tortuosity inside the EMs².

The 3D geometry “CFD” module was used to solve the flow velocity and streamline distribution in the simulated pores. The range of HRT in the simulation (3–182 s) was consistent with the experiment (3–182 s). The Reynolds number (Re) is defined by Supplementary equation (4)^{3,4},

$$Re = \frac{\rho U d_p}{\mu} \quad (4)$$

where U is the flow velocity (m s^{-1}), d_p is the diameter of the pore (m), and μ is the dynamic viscosity ($\text{kg m}^{-1} \text{s}^{-1}$). Since the Re remains below 1 even at the highest flow velocities on EMs (Supplementary Table 5), the flow in the microchannels was confirmed to be laminar. The solution’s inflow/outflow was modeled as steady-state

flow (transient velocity fluctuations were not accounted for).

The properties of the electrode and electrolyte are assumed homogeneous throughout the pore. The electrode was treated as highly conductive, neglecting internal IR drop, and therefore it is not affected by the limiting current density (i.e., the local maximum current density in the electrode).

The properties of the electrode and electrolyte are assumed homogeneous throughout the pore. The electrode was treated as highly conductive, neglecting internal IR drop, and therefore it is not affected by the limiting current density (i.e., the local maximum current density in the electrode). The “transport of diluted species” module was used to solve the concentration distribution. The mathematical model of the mass transfer process includes Fick’s law, the convection process, and the reaction process. A 100 mM Na₂SO₄ electrolyte was used in the experiment, allowing us to ignore the electrostatic migration of NO₃⁻ and NH₄⁺, thus simplifying the model⁵. The disturbance of gas evolution on flow and mass transfer was neglected. The channels were assumed to be fully electrolyte-filled, with no gas blockage. The convection–diffusion equation for each species was given by Supplementary equation (5)⁶,

$$R_i = -\nabla \cdot [D_i \nabla C_i + C_i \mathbf{u}] \quad (5)$$

where C_i is the concentration of species i , R_i is the reaction flux of species i , and D_i is the diffusion coefficient of species i .”

Page 5 in the revised Supplementary Information:

“The overpotential (η) was calculated by Supplementary equation (7)¹¹,

$$\eta = \varphi_m - \Delta\varphi_s - \varphi_e \quad (7)$$

where φ_m is the electrode potential (V), $\Delta\varphi_s$ is the solution potential drop (V), and φ_e is the equilibrium potential (V).”

The parameter determination process is supplemented as follows:

Page 50 in the revised Supplementary Information:

“**Supplementary Table 5.** Estimated flow velocity and Re of EMs.

Highest flow velocity ^a	Re
------

	[cm s ⁻¹]	
EM₄	0.88	0.04
EM₇	0.55	0.04
EM₂₄	0.38	0.09
EM₅₈	0.28	0.16
EM₈₀	0.25	0.20

^aThe highest flow velocity in the pores was calculated based on the highest flow rate and porosity observed in the experiments.”

Page 51 in the revised Supplementary Information:

“**Supplementary Table 6.** Parameters and values used in the simulation.

Parameters	Value
Hydrogen evolution potential (E_H) ²⁵	0 [V]
Exchange current density for hydrogen evolution (i_H) ^a	1×10^{-2} [A m ⁻²]
Charge transfer coefficient for hydrogen evolution (α_H) ^{26 a}	0.75
Nitrate reduction potential (E_{NH}) ²⁷	0.69 [V]
Exchange current density for nitrate reduction (i_{NH}) ^a	$5.7 \times 10^{-10} C_O^{*(1-\alpha)} C_R^{*\alpha}$ [A m ⁻²]
Charge transfer coefficient for nitrate reduction (α_{NH}) ^a	0.6
Current of one simulated pore (i_{app}) ^b	-5.6×10^{-6} , -3.4×10^{-6} , -7.7×10^{-7} , -1.0×10^{-7} , -5×10^{-8} [A]
Inlet concentration ^c	1×10^{-4} , 1×10^{-3} , 1×10^{-2} [M]
HRT for nitrate reduction ^c	6.8 – 181.7 [s]
Electrolyte conductivity ^c	0.256, 1.28, 6.4 [S m ⁻¹]
Diffusion coefficient ^d	1×10^{-9} [m ² s ⁻¹]
Tortuosity of pore model (τ) ^e	1.4

^aDue to the three-dimensional structure of the EMs, these values were optimized in the simulation. The catalytic performance at different currents, fluxes, electrolyte

concentrations and electrode thicknesses was also simulated to check the validity of the fitted model parameters. The influence of these optimized parameters on the simulation results is discussed in the Discussion section and Supplementary Fig 39.

^bCurrent of EM₈₀, EM₅₈, EM₂₄, EM₇, and EM₄ in one simulated pore was determined according to Supplementary equation (5), respectively.

^cThese values were determined by the experimental operating conditions.

^dThe diffusion coefficient of nitrate ions is around $1.5 \times 10^{-9} \text{ m}^2 \text{ s}^{-1}$ ²⁸; however, since the model did not incorporate factors such as pore connectivity or bubble-induced mass transfer effects, a value of $1 \times 10^{-9} \text{ m}^2 \text{ s}^{-1}$ was adopted in the simulation. The influence of this parameter on the simulation results is discussed in the Discussion section and Supplementary Fig. 39.

^eThe tortuosity of porous titanium foam ranges from 1.2 to 1.8², and a value of 1.4 was selected for the simulation.”

The influence of parameters is supplemented as follows:

Line 496 – 505 in the revised manuscript:

“The impact of other key design parameters, including applied current, intrinsic catalytic activity, reactant diffusion coefficient, and electrode spacing, were also analyzed by simulation, respectively (Supplementary Fig. 39). As the applied current increases, faster mass transfer is required to balance electron transfer, which amplifies the advantage of smaller pores and may reduce the optimal pore size. Meanwhile, improvements in intrinsic catalytic activity enhance surface reaction rates but have minimal influence on the optimal pore size. Similarly, variations in diffusion coefficient and electrode spacing show little effect on the optimal pore dimension. This in-depth understanding of the structure-performance relationship offers new insights into the design of EMs with ultrahigh reaction kinetics, advancing beyond conventional catalyst development paradigms.”

Page 45 in the revised Supplementary Information:

“

Supplementary Fig. 39. Simulated reaction kinetics of Sim4, Sim7, and Sim24 under varied conditions: **(a)** applied current, **(b)** reactant diffusion coefficient, **(c)** hydrogen evolution reaction exchange current density, **(d)** nitrate reduction exchange current density, and **(e)** electrode spacing. The y-axis represents the ratio of k_{obs} of sim4 or sim24 to the k_{obs} of Sim7 under an identical condition. $D = 1 \times 10^{-9} \text{ m}^2 \text{ s}^{-1}$.”

6. The reduction of TCAA produces MCAA as an intermediate (Fig. 3b), which is reported to be more toxic than TCAA (Environ. Sci. Technol. 2000, 34 (13), 2675-2683). The capability of EMs to achieve complete dechlorination should be further investigated and discussed.

Thank you for your suggestion. In our initial experiment (Fig. 3b), both MCAA and AA were the primary products, due to insufficient reaction time. We have supplemented the data collected under extended reaction time. When the hydraulic residence time (HRT) was prolonged to 182 seconds on EM₇, the MCAA concentration decreased to 0.06 mM, with AA becoming the dominant product. At this point, over 97% of the chlorine was successfully removed, demonstrating that EM₇ can achieve complete dechlorination by extending the HRT. Experimental results on other pore-sized EMs under extended HRTs have also been added to the Supplementary Information (Supplementary Fig. 19).

We have added the relevant statements in the revised main text and revised Supplementary Information.

Line 257 – 265 in the revised manuscript:

“With an increase in HRT, the TCAA concentration decreased rapidly, while the concentration of DCAA and MCAA increased and peaked at 0.36 mM and 0.31 mM, respectively (Fig. 3b). Further increasing the HRT led to a decline in DCAA and MCAA concentrations, indicating their roles as intermediates in subsequent hydrodechlorination reactions. More than 92% of TCAA was converted into AA as the final product at an HRT of 182 s, corresponding to a total dechlorination efficiency of 94% (Fig. 3c). In contrast, the dechlorination efficiency on EM₈₀ was less than 80% over the same duration, with both MCAA and AA as the main products (Supplementary Fig. 19). As a result, EM₇ consistently achieved significantly higher FEs than the other EMs (Supplementary Fig. 20).”

Line 290 – 293 in the revised manuscript:

Fig. 3 TCAA dichlorination and phenol hydrogenation performance on EMs. a Removal of TCAA on the EMs. **b** Concentration evolution of TCAA and its dichlorination products on EM₇.”

“Supplementary Fig. 19. Concentration evolution of TCAA and its dechlorination products on (a) EM₄, (b) EM₂₄, (c) EM₅₈, and (d) EM₈₀, respectively. Electrolyte: 0.1 M Na₂SO₄ (pH = 7).”

7. The stability of the electrodes is a critical issue. The authors should provide data on Ru leaching during electrolysis to validate the long-term performance and robustness of the electrodes.

We sincerely appreciate your valuable suggestion. ICP-MS analysis of the effluent from the nitrate reduction stability test revealed that the Ru concentration remained consistently below the detection limit (0.1 ppb). This demonstrates the superior stability of the atomically dispersed Ru species under reductive conditions, which aligns well with existing research findings (*J. Am. Chem. Soc.* **2022**, 144, 20834–20846; *Environ. Sci. Technol.* **2022**, 56, 7, 4356–4366).

We have incorporated the relevant statements in the main text.

Line 224 – 226 in the revised manuscript:

“The Ru concentration in the effluent consistently remained below the detection limit

(0.1 ppb) of inductively coupled plasma mass spectrometry (ICP-MS), indicating negligible Ru leaching.”

8. The water flow directions depicted in Figs. 1a and 2a appear inconsistent. Revising these diagrams for clarity is recommended.

We appreciate your suggestion. In our experimental setup, the solution flows sequentially from the bottom through the cathode and then the anode, which facilitates bubble removal. We have updated the liquid flow direction in Fig. 1a accordingly.

Figure 1a in the revised manuscript:

Fig. 1 Schematic and characterization of EMs. a Schematic of the spatial confinement effect on EMs.”

Line 535 – 537 in the revised manuscript:

“The solution flows sequentially from the bottom through the working electrode (EM) and then the counter electrode, which facilitates bubble removal.”

9. Supplementing Fig. 3d with the reaction equations for phenol reduction would enhance the clarity of the results.

Thank you for your suggestion. Based on GC-MS analysis of the products, phenol first undergoes a 4-electron reduction to cyclohexanone (intermediate), followed by a 2-electron reduction to cyclohexanol. The corresponding reaction equations have been added in Figure. 3d.

Figure 3d in the revised manuscript:

Fig. 3 TCAA dichlorination and phenol hydrogenation performance on EMs. d Removal of phenol on the EMs.”

Line 277 – 278 in the revised manuscript:

“Phenol first undergoes a 4-electron reduction to cyclohexanone (intermediate), followed by a 2-electron reduction to cyclohexanol.”

10. The explanation of reaction distribution in the microchannels (Lines 371–377) is somewhat ambiguous. The authors should revise these sentences for improved clarity and precision.

We appreciate your valuable comments, which significantly increased the readability of our manuscript. In lines 371-377, we primarily focused on the issue of nonuniform reaction distribution arising from uneven current distribution within the pore channels. To improve clarity, we have supplemented the average concentration in the cross-sections. By comparing two cross-sections at $x = 40 \mu\text{m}$ and $x = 10 \mu\text{m}$, Sim₇ exhibited a concentration reduction of 0.13 mM (from 0.34 mM to 0.21 mM), while Sim₂₄ showed a smaller decrease of 0.1 mM (from 0.33 mM to 0.23 mM).

Notably, Sim₇ displayed higher concentrations than Sim₂₄ at $x = 40 \mu\text{m}$ but lower concentrations at $x = 10 \mu\text{m}$, indicating that reactions become more surface-concentrated in smaller pore channels. Although Sim₄ exhibits a greater concentration reduction of 0.15 mM (from 0.44 mM to 0.29 mM), its effluent concentration remains higher than Sim₇ because the reaction is overly concentrated on the surface (Fig. 5c).

We have revised the relevant statements in the main text.

Line 427 – 434 in the revised manuscript:

“It is also clearly shown that the nitrate reduction reaction primarily occurs near the pore opening. The smaller the pore size, the more unevenly the reaction is distributed along the channel. By comparing the average concentrations at two cross-sections ($x = 40 \mu\text{m}$ and $x = 10 \mu\text{m}$), Sim₄ exhibited a concentration drop of 0.15 mM (from 0.44 mM to 0.29 mM), whereas Sim₈₀ showed a smaller decrease of 0.08 mM (from 0.40 mM to 0.32 mM). Notably, although Sim₄ exhibits a larger local concentration reduction, its effluent concentration remains higher than that of Sim₇ due to excessive surface-localized reactions (Supplementary Fig. 32).”

Figure 5a in the revised manuscript:

Fig. 5 Multiphysics simulation results of nitrate reduction on EMs. a Visualized nitrate concentration in the cross-sections ($x = 10$ and $40 \mu\text{m}$) of the EM channels at an HRT of 55 s.”

Reviewer #2 (Remarks to the Author):

This study prepares a range of electrified membranes with varying pore sizes including atomically dispersed Ru to demonstrate reduction of a variety of pollutants, including nitrate, trichloroacetic acid, and phenol. They obtain a semi-volcano-shaped relationship between the electrocatalytic activity and the pore size, where the smallest pore size tested shows a decrease (different trend from the rest). While this study does contribute new data to the literature on flow-through electrode pore sizes and conversion trends across different contaminants, it is not clear that it provides any new fundamental insight. A major concern is that the explanation for the trend (and the modeling conducted) assumes that current varies through the depth of the membrane, but this should not be the case for a uniformly conductive monolithic electrified membrane (i.e., electrons should transfer freely throughout the conductive solid, and the current at any given point in the membrane should be the same). There are also many other potential issues in the choice of experimental parameters measured that are used to draw conclusions, as detailed below. It is possible that the trend is due to optimization of transport rates in a reactive system, as previously reported. I believe that it is important for the authors to consider and address these points detailed below to see if their trends and explanations still hold after these corrections.

Thank you for your suggestions. We would like to clarify that the varied current through the depth of the membrane is not an assumption. Actually, it is a result of rigorous deduction, verified by both experimental and theoretical analysis. It is also consistent with findings from other researches (*Environ. Sci. Technol.* **2022**, 56, 9722–9731; *Electrochimica Acta.* **1996**, 41, 519–526). Regarding your main concern, we have made three key additions to the article:

- (1) We have added direct experimental observations confirming that different locations on the electrode exhibit varying current densities due to their differing distances from the counter electrode. This phenomenon occurs even with highly conductive electrodes (such as the Pt electrode used in our observation experiment).
- (2) We have provided a detailed mechanistic explanation for the observed nonuniform current distribution, supported by a newly added schematic diagram (Fig. 4d). Current and potential distributions in porous electrodes were analyzed using the well-established transmission line model (*J. Electrochem. Soc.* **1978**, 125, 58; *Small Methods* **2018**, 2, 1700342; *Environ. Sci. Technol.* **2022**, 56, 9722–9731).
- (3) We significantly expanded the modeling assumptions and implementation details. In our simulations, the local current density within the pores is calculated from the

local overpotential using the Butler-Volmer equation (Bard, A. J., Faulkner, L. R. *Electrochemical Methods: Fundamentals and Applications*). Under the assumptions of high electrode conductivity and uniformity, the simulation results show good agreement with experimental observations.

A detailed point-by-point response to your comments is as follows.

MAJOR COMMENTS

1. Lines 86-89, TiO₂ is described as being a conductive substrate, but it's insulating. The authors need to be clearer about the electrical conductivity of the substrate, which will have significant implications for their major hypothesis / finding that the current varies through the depth of the membrane substrate.

Thank you for your comment. Actually, our results indicate that the nonuniform current is not (at least not directly) attributable to the semiconducting TiO₂ in the EMs. On one hand, TiO₂ mainly serves as an interlayer on the metallic Ti substrate to facilitate the uniform loading of Ru catalyst. In this study, the thin TiO₂ layer is highly doped with oxygen vacancies, which significantly increases its conductivity. On the other hand, even on a highly conductive metal such as Pt, the current varies through the depth of the electrode. Our detailed explanations can be found below.

(1) Conductivity of the substrate

Pure TiO₂ is indeed an n-type semiconductor and exhibited rectification effect. As a result, TiO₂ can be hardly used as an anode because the oxidation reaction on its surface can only take place with the involvement of the minority carriers (i.e., holes, details can be found in Memming, R., *Semiconductor Electrochemistry*. WILEY-VCH Verlag GmbH: 2015 and *Chem. Rev.* **2012**, 112 (10), 5520-5551). On the contrary, it can be used as a cathode because the reduction reactions can proceed with the aid of majority carriers (electrons, see *ACS Catal.*, **2018**, 8, 4288).

In our study, only a thin layer (less than 5 μm) of TiO₂ nanosheets is coated on the substrate of metallic Ti, which exhibits high conductivity. Additionally, a hydrogen annealing treatment was conducted to fully reduce TiO₂, which can significantly increase free electron concentrations through the introduction of oxygen vacancies (*Chem. Soc. Rev.* **2015**, 44 (7), 1861-1885). **As a result, all the EMs exhibited superior conductivity, with the resistances below 2 Ω , as confirmed by the electrochemical impedance spectroscopy results (Supplementary Fig. 9).** This

confirms that the TiO₂ semiconducting thin film does not significantly compromise the electrode's conductivity.

(2) Current distribution of highly conductive electrode

Furthermore, although all EMs exhibit good conductivity, their high conductivity does not negate the conclusion regarding nonuniform current distribution. Although monolithic electrodes with uniform conductivity are equipotential by nature, their porous structure is filled with electrolytes, which is essential for electrochemical reactions. Within such pores, an IR drop of the electrolyte occurs at varying depths, leading to differences in solution potential drop ($\Delta\phi_s$) across the pore structure. Variations in $\Delta\phi_s$ result in differing local overpotentials (η) (calculated as $\eta = \phi_m - \Delta\phi_s - \phi_e$, where ϕ_m is the local electrode potential and ϕ_e is the equilibrium potential, *Electrochimica Acta*. **1996**, 41, 519–526.). This overpotential gradient ultimately drives the nonuniform current distribution within the electrode. A more detailed explanation of the nonuniform current distribution, along with supporting schematic diagrams, has been incorporated into the main text.

Additionally, the current density distribution at the interface of a highly conductive Pt electrode was characterized with the aid of a laser scanning confocal microscopy and an OH⁻ fluorescent probe. To emulate the configuration of an EM, the Pt electrode was positioned perpendicular to the counter electrode. Supplementary Fig. 23 shows the spatial OH⁻ distribution at the cathode, which directly reflects local current density variations. As the reaction proceeded, the generated OH⁻ diffused outward from the electrode surface, forming a clear fluorescent boundary. The near end of the electrode produced OH⁻ at the fastest rate. The closer to the counter electrode, the more OH⁻ is produced. This experiment clearly demonstrates the nonuniform current distribution of highly conductive electrode.

We also conducted an optical observation experiment to assess the impact of electrolyte concentration on current distribution (Supplementary Fig. 28). At high electrolyte concentration (300 mM), OH⁻ ions were generated uniformly across the entire Pt electrode surface (indicated by dark regions). However, when the electrolyte concentration was reduced to 2 mM, OH⁻ production became spatially localized at the region near the counter electrode. This observation confirms that the variations in $\Delta\phi_s$ are the primary driver of nonuniform current distribution on highly conductive electrode surfaces.

The fluorescence and optical observation experiments collectively confirm that current unevenness occurs across different spatial scales. These observations align with

the experimental results presented in our manuscript and are also consistent to the previously reported results (*Environ. Sci. Technol.* **2022**, 56, 9722–9731; *ACS ES&T Engineering* **2023** 3 (12), 2194-2201). Notably, the current nonuniformity observed in the nitrate reduction experiments is expected to be more pronounced than in the aforementioned visualization experiments. This is because the pollutant removal experiments were performed at higher currents, resulting in greater IR drop and consequently more pronounced current nonuniformity. We have added the relevant statements in the main text and Supplementary Information.

Line 125 – 126 in the revised manuscript:

“All the EMs demonstrated extremely low series resistance values, below 2Ω (Supplementary Fig. 9), confirming the high conductivity of the defective TiO₂ nanosheets³⁷.”

Line 316 – 321 in the revised manuscript:

“This nonuniformity trend is also visually evidenced by an OH⁻ fluorescence probe experiment at the Pt electrode interface (Supplementary Fig. 23 and Movie 1). The formation of the fluorescent boundary at the rear side of the electrode occurred significantly more slowly than at the front, highlighting the spatial variation in local electrochemical activity. Notably, such nonuniformity of current distribution is more pronounced for channels with smaller pores (Fig. 4b).”

Line 345 – 362 in the revised manuscript:

“Based on the well-established transmission line model, the nonuniform current distribution is primarily attributed to the solution resistance (R_s), which is influenced by both the pore size and electrolyte conductivity (Fig. 4d)^{47,48}. As the distance from the counter electrode increases, the cumulative R_s grows, leading to a larger solution potential drop ($\Delta\phi_s$). Although the electrode potential (ϕ_m) remains constant throughout the depth, this difference in $\Delta\phi_s$ results in variations in the local overpotentials (η), calculated as $\eta = \phi_m - \Delta\phi_s - \phi_e$, where ϕ_e represents the equilibrium potential⁴⁹. This spatial variation in overpotential ultimately causes the observed nonuniform current distribution within the EMs. Considering the critical role of electrolyte concentration on R_s , its impact on the current distribution was quantitatively simulated (Supplementary Fig. 25). It is observed that increasing the electrolyte concentration significantly reduces the ratio of current at the nearest end to that at the farthest end (Supplementary Fig. 26). For instance, the near-to-far current ratio of Sim₈₀ decreased

from 6.4×10^3 to 42 when the electrolyte concentration was increased from 0.02 M to 0.5 M. Similarly, the current ratio of Sim₄ also underwent a drastic decrease from 3.8×10^5 to 4.0×10^2 . As the electrolyte concentration increases, the charge transfer of nitrate occurs throughout the entire length of the channel (Supplementary Fig. 27), indicating an expansion of the reaction zone. This phenomenon can be attributed to the reduced R_s , which decreases the potential drop ($\Delta\phi_s$) across the pores.”

Line 365 – 374 in the revised manuscript:

“Upon increasing the electrolyte concentration, the Cu deposition depth was significantly enhanced. For instance, the Cu deposition depth on EM₇ in 0.1 M Na₂SO₄ was only 0.6 mm, which is substantially lower than that of EM₈₀ (1.6 mm). However, when the electrolyte concentration was raised to 0.5 M, the deposition depth on EM₇ increased markedly to 2.3 mm. To further validate the influence of electrolyte concentration on current density distribution, an optical observation experiment was performed using a Pt electrode and a pH-sensitive probe to visualize the electrogenerated OH⁻ ions (Supplementary Fig. 28 and Movie 2). At a high electrolyte concentration of 0.3 M, OH⁻ ions were generated uniformly across the entire Pt electrode surface. In contrast, under low electrolyte concentration conditions (0.002 M), OH⁻ production became highly localized in the region near the counter electrode.”

Figure 4d in the revised manuscript:

d Equivalent circuit diagram and schematic illustration of potential distribution within EM pores, where R_s represents the distributed resistance of the electrolyte, R_F represents the Faradic charge transfer resistance, and Z_w is the Warburg impedance.”

Page 15 in the revised Supplementary Information:

“**Supplementary Fig. 9.** Electrochemical impedance spectroscopy of EMs (inset: magnified view illustrating the x-intercept).”

Page 29 in the revised Supplementary Information:

“

Supplementary Fig. 23. Fluorescence observation of current density distribution on Pt electrode surface. (a) Photograph of electrode configuration. (b) Evolution of pH

distribution at electrode interface. The Pt electrode was positioned perpendicular to the counter electrode to simulate the EM. Due to instrument limitations, the upper and lower parts of the electrode were recorded in two separate repeat experiments. BCECF-AM was employed as OH^- probe in solution. Current: 0.5 mA, electrolyte: 0.02 M Na_2SO_4 .”

Page 34 in the revised Supplementary Information:

Supplementary Fig. 28. Optical observation of current density distribution on the Pt electrode surface across electrolyte concentrations ranging from 2 mM to 300mM. (a) Photograph of electrode configuration. (b) Evolution of pH distribution at electrode interface. Bromothymol blue was employed as a pH probe and the same setup as in the fluorescence observation experiment was used. Current: 0.6 mA.”

2. Especially since the pore size of the EMs is so important, more information about the data collected on determining pore size, including standard deviation or a pore size distribution, should be provided.

Thank you for your suggestions. We characterized the pore size distribution of the EMs using mercury intrusion porosimetry (Fig. 1g), revealing that each EM exhibited a standard monomodal pore size distribution. Based on this, the nominal pore size of the EMs was determined by the modal pore diameter (i.e., pore diameter of the maximum in a differential pore size distribution curve).

To further quantify the pore size characteristics, we fitted the pore size distribution data of each EM to a log-normal distribution (C. Causserand, P. Aimar *Comprehensive Membrane Science and Engineering*), using the modal pore size as the mathematical expectation. Based on the fitting results, we determined the standard deviations of the pore sizes for each EM, which were 3.8 ± 0.4 , 7.3 ± 1.4 , 24.3 ± 1.6 , 58.6 ± 10.5 , and 80.5 ± 14.6 μm , respectively. These data provide quantitative evidence that the study successfully fabricated four EMs with controlled pore sizes spanning 4 to 80 μm .

We have added the relevant statements in the main text and Supplementary Information.

Line 106 – 110 in the revised manuscript:

“The Ru/TiO₂ EMs with different pore sizes were fabricated. As determined by the mercury intrusion method, all EMs exhibited a standard monomodal pore size distribution, with modal pore diameters ranging from 4 to 80 μm (Fig. 1g). According to the log-normal distribution fitting results (Supplementary Fig. 6)³⁶, the standard deviations of the pore sizes for each EM were 3.8 ± 0.4 , 7.3 ± 1.4 , 24.3 ± 1.6 , 58.6 ± 10.5 , and 80.5 ± 14.6 μm , respectively.”

Page 12 in the revised Supplementary Information:

Supplementary Fig. 6. Log-normal fitting curves of the pore size distributions for (a) EM₄, (b) EM₇, (c) EM₂₄, (d) EM₅₈, and (e) EM₈₀”

3. P. 5-6, the authors discuss the lack of correspondence between BET surface area and LSVs. However, they should quantify the electrochemically active surface area, which they should be able to do from their cyclic voltammetry. This could eliminate the lack of correspondence that they observe, since there is a correlated trend.

Thank you for your suggestion. As you pointed out, electrochemical surface area (ECSA) is indeed more appropriate for comparison with linear sweep voltammetry

(LSV) because it directly reflects electrochemical activity. Based on cyclic voltammetry data (Supplementary Fig. 7), we calculated the double-layer capacitance (C_{dl}) for EMs with different pore sizes, which is proportional to ECSA. As the pore size decreased, the C_{dl} of the EM increased, showing a trend generally consistent with BET measurements. Notably, EM₄ exhibited a higher C_{dl} than EM₇, which still does not align with the LSV results.

This discrepancy can be explained as follows. While both ECSA and LSV are related to electrochemical active surface area, they differ in sensitivity to nonuniform current distribution within the pores. As discussed in our **Response to Comment #1**, the negligible IR drop under the low-current conditions in ECSA measurements ($\leq 150 \mu\text{A cm}^{-2}$) allows for relatively uniform reaction distribution (ion adsorption/desorption) throughout the EM's pore structure. In contrast, LSV operates at significantly higher currents ($\leq 150 \text{ mA cm}^{-2}$) where substantial IR drop causes severe reaction zone confinement within EMs. Moreover, smaller pores contain reduced solution volume, leading to higher solution resistance, which makes the nonuniform distribution more pronounced in small-pore EMs.

We have revised the relevant statements in the main text and Supplementary Information.

Line 120 – 125 in the revised manuscript:

“The electrochemical double-layer capacitance (C_{dl}), directly related to the electrochemically active surface area (ECSA), was determined via cyclic voltammetry (Supplementary Fig. 7). A comparison of C_{dl} values revealed that the ECSA increased as the pore size decreased and EM₄ exhibited the largest effective surface area, which is consistent with the Brunauer–Emmett–Teller (BET) analysis (Supplementary Fig. 8 and Supplementary Table 1).”

Line 127 – 134 in the revised manuscript:

“To further assess their electrocatalytic activity, linear sweep voltammetry (LSV) was performed (Fig. 1i). While the current densities generally increased with surface area, they did not scale proportionally. For instance, although EM₄ exhibited an ECSA 2.8 times that of EM₈₀ (Supplementary Table 1), its cathodic current density at $-0.2 V_{RHE}$ (56 mA cm^{-2}) was only 51% higher than that of EM₈₀ (37 mA cm^{-2}). Notably, EM₇ exhibited a current even higher than that of EM₄, despite possessing a smaller ECSA. This discrepancy suggests the existence of electrochemical nonuniformity within the micropores at relatively high current densities, where the full ECSA cannot be

efficiently utilized³¹.”

Page 13 in the revised Supplementary Information:

Supplementary Fig. 7. (f) C_{dl} by plotting current variation against the scan rate to fit a linear regression. C_{dl} values were measured at the potential of 0.91 V vs. RHE. Electrolyte: 0.1 M Na_2SO_4 .”

Page 46 in the revised Supplementary Information:

“**Supplementary Table 1.** The surface area and porosity of EMs.

	BET Area	Mercury Intrusion	C_{dl}	Porosity ^b
	$[\text{m}^2 \text{g}^{-1}]$	Area ^a	$[\text{mF cm}^{-2}]$	$[\%]$
		$[\text{m}^2 \text{g}^{-1}]$		
EM ₄	10.90	3.23	4.44	10
EM ₇	6.80	1.92	3.66	16
EM ₂₄	3.47	1.76	2.85	23
EM ₅₈	2.30	0.75	2.16	31
EM ₈₀	2.08	0.53	1.60	35

^aThe mercury intrusion area is obtained from the cumulative specific surface area when the pore size is greater than 21 nm.

^bThe porosity of EMs was determined by Archimedes’ Drainage Method.”

a. The authors argue that differences in surface area in the membranes with different pore sizes cannot account for the differences in conversion activity (specifically, the

trends are similar, but not exactly proportional). However, they use BET surface area as the relevant parameter. This is a measure of physical gas adsorption sites, but the relevant surface area is related to electrochemical activity. Therefore, electrochemically active surface area should be used as the relevant parameter to confirm whether these changes account for the differences and performance for the membranes with different pore sizes.

We appreciate your suggestion. In our **Response to Comment #3**, we supplemented the analysis of LSV current based on ECSA area, which cannot fully explain the differences in reduction current and performance among EMs.

Additionally, we conducted supplementary ECSA tests on EM₇ with varying thicknesses. The results revealed a gradual decrease in ECSA as EM thickness reduces, attributed to the reduced electrochemical surface area in thinner EMs. However, our experimental results (Fig. 4h) demonstrate that EM₇ with different thicknesses (and consequently different ECSA values) exhibited consistent catalytic performance. This finding reinforces the conclusion that the performance variations among EMs are likely governed by mass transport and current distribution effects rather than surface area alone.

Line 391 – 400 in the revised manuscript:

“It can be found that the thickness of EM₇ had little impact on its catalytic performance (Fig. 4h and Supplementary Fig. 30). Even when the thickness (and consequently the Ru loading) was reduced to one-third, the 1mm-EM₇ still exhibited an extraordinary nitrate conversion rate. By comparing the C_{dl} values of EM₇ with different thicknesses (Supplementary Fig. 31), it is evident that the ECSA of 1mm-EM₇ is significantly lower than that of 3mm-EM₇. The discrepancy indicates that the ECSA cannot be fully utilized in microchannels with relatively high current densities and low solution conductivities. These findings suggest that EM₇ of varying thicknesses share a similar electroactive thickness, highlighting the potential to maintain high catalytic performance while reducing material usage and fabrication costs.”

Figure 4h in the revised manuscript:

Fig. 4 Current distribution in the channel of EMs. h Nitrate removal performance on EM₇ with different thicknesses.”

Page 37 in the revised Supplementary Information:

Supplementary Fig. 31. The cyclic voltammetry (CV) curves of EMs with different pore sizes and the corresponding electrical double-layer capacitance (C_{dl}). CV curves of (a) 2mm-EM₇, (b) 1mm-EM₇, and (c) C_{dl} of EM₇ with different thicknesses by plotting current variation against the scan rate to fit a linear regression. C_{dl} values were measured at the potential of 0.91 V vs. RHE. Electrolyte: 0.1 M Na₂SO₄.”

4. The authors do not seem to report any potential reference to a standard reference electrode. This will make the results impossible to compare with other nitrate reduction results in existing literature, or to have a sense of how large the potential and overpotential are.

We appreciate your suggestion. The potential referenced to the standard reference electrode is essential for comparison with other studies. Therefore, we measured the potentials of the EMs with the aid of an Ag/AgCl reference electrode and converted the values into the reversible hydrogen electrode (RHE) scale (V_{RHE}). Under an identical current density (39.3 mA cm^{-2}), the working potential fell in a small range of -0.22 to $-0.25 V_{RHE}$ across all EMs. We also optimized the placement of the reference electrode (within 1 mm of the working electrode) to minimize the influence of solution resistance on potential measurements. Accordingly, we have updated the LSV curves, which maintain the same overall trend as before.

In addition, we have included LSV curves for EM₇ under varying NO₃⁻ concentrations (Supplementary Fig. 16). A comparative analysis of these results with prior studies on nitrate reduction current demonstrates that our catalyst exhibits excellent activity for nitrate reduction.

Line 149-150 in the revised manuscript:

“The operating potentials of the EMs were nearly identical, falling within a narrow range of -0.22 to $-0.25 V_{RHE}$ (Supplementary Table 2).”

Line 189 – 195 in the revised manuscript:

“LSV tests under varying nitrate concentrations showed a marked increase in reduction current with increasing nitrate levels (Supplementary Fig. 16a). Notably, EM₇ exhibited significantly higher reduction currents than many previously reported electrocatalysts, even at relatively lower nitrate concentrations (Supplementary Fig. 16b and Table 3), highlighting its superior electrocatalytic activity. These results suggest that DET is the

predominant reduction pathway, with no significant variation observed among the different EMs.”

Page 22 in the revised Supplementary Information:

Supplementary Fig. 16. (a) Linear sweep voltammetry (LSV) curves of EM₇ in the presence of different concentrations of nitrate. (b) Comparison of the nitrate reduction current on EM₇ with that of other electrocatalysts. Details included in Supplementary Table 3. Electrolyte: 0.1 M Na₂SO₄ (pH = 7).”

Page 47 in the revised Supplementary Information:

“Supplementary Table 2. The operating potential of the EMs.^a

	Potential [V _{RHE}]
EM ₄	-0.220 ± 0.003
EM ₇	-0.212 ± 0.006
EM ₂₄	-0.236 ± 0.003
EM ₅₈	-0.238 ± 0.004
EM ₈₀	-0.249 ± 0.004

^aThe potential was measured under conditions of 0.1 M Na₂SO₄ with 1 mM NaNO₃, at an HRT of 7 s.”

Page 48 in the revised Supplementary Information:

“Supplementary Table 3. Detailed comparison of nitrate reduction current.^a

Catalyst	Electrolyte	Potential [V _{RHE}]	Current density [mA cm ⁻²]	Ref
Ni ₁ Cu-SAA	0.5 M K ₂ SO ₄ 200 ppm NO ₃ ⁻ -N	-0.2	-20	17
IrNiCu@Cu-20	0.1 M KOH 0.1 M KNO ₃	-0.1	-82	18
Co NC/Graphene	1 M KOH 1 M KNO ₃	-0.2	-122	19
CoP-CNS	1 M OH ⁻¹ 1 M NO ₃ ⁻	-0.2	-102	20
Pd-CuO-200	1 M KOH 0.1 M KNO ₃	-0.2, -0.1	-183, -139	21
Cu-NBs-100	1 M KOH 0.1 M KNO ₃	-0.2	-350	22
Fe/Cu-HNG	1 M KOH 0.1 M KNO ₃	-0.2	-15	23
Cu ₅₀ Ni ₅₀	1 M KOH 0.1 M KNO ₃	-0.1	-10	24

^aAll data used for comparison are derived from LSV measurements.”

5. Line 188, the HRT of seven seconds is used for the energy consumption calculation, but 80% nitrate conversion is assumed, which seems to require an HRT around 55 seconds. The considered conditions do not seem to be consistent for the parameters used in these calculations.

We appreciate your suggestions and apologize for any ambiguities in the original manuscript. In the initial version, we compared transmembrane pressures across different pore-sized EMs using pressure data at an HRT of 7 s. However, the pumping energy consumption calculations were based on the HRT values where 80% nitrate removal is achieved (e.g., 35 s for EM₇, 85 s for EM₈₀).

To improve clarity, both transmembrane pressure and energy consumption referred in the manuscript were aligned with a single-pass 80% nitrate removal benchmark. We have revised the relevant statements in the main text and added the relevant calculation method in the Supplementary Information.

Line 212 – 218 in the revised manuscript:

“The energy consumption of the EMs required to achieve 80% nitrate removal efficiency was calculated (Fig. 2g and Supplementary Table 4). Notably, EM₇ and EM₈₀ required HRT of 35 s and 85 s, respectively, to reach this target. Under these conditions, the transmembrane pressure on EM₇ reached 6.0 kPa (Supplementary Fig. 18), which is 8.5 times higher than that of EM₈₀ (0.7 kPa). However, the contribution of pumping energy per unit of nitrate removed is negligible compared to electrical energy. For instance, the pumping energy for EM₇ was 0.003 kWh g-N⁻¹, accounting for only 0.7% of the total energy consumption (0.39 kWh g-N⁻¹).”

Page 3 in the revised Supplementary Information:

“The hydraulic residence time (HRT) required to achieve an 80% removal rate was calculated using Supplementary equation (1),

$$\text{HRT}_{80\%} = \frac{\ln(5)}{k_{\text{obs}}} \quad (1)$$

”

6. Lines 279-293, it is not clear why current would change with the depth of the channels. If the membrane is conductive, and if the material composition and therefore conductivity is uniform throughout, then the potential at any given point in the membrane should be the same as the potential at any other point. It seems as though the authors are treating the thickness of the membrane as if it were an electrolyte, which is vanishingly conductive to electrons, rather than as an electrode. However, electrodes are typically highly electrically conductive, meaning that delocalized electrons move freely and rapidly throughout the conductive material (by definition, an electrode is the conductive interface that allows free electron transfer). Therefore, the entire concept of current changing with the depth of the membrane, and therefore the explanation about why differences are observed for membranes with different pore sizes, does not seem to make sense. This is very important here: the key novel scientific ideas in this manuscript seem to rest on this idea.

Thank you for your valuable comment, as our paper lacked a detailed explanation on nonuniform current distribution. On the one hand, as you pointed out, the electrode is highly conductive and has a uniform material composition, ensuring consistent internal potential (φ_m). On the other hand, although we did not treat the electrode as an electrolyte, it is indeed filled with electrolyte solution. Within the porous structure, the IR drop (which is often neglected in experiments with flat electrodes) gives rise to different electrolyte potential drops ($\Delta\varphi_s$), leading to depth-dependent overpotential ($\eta = \varphi_m - \Delta\varphi_s - \varphi_e$) for reaction. This is the fundamental driving current density gradients. As for EMs with different pore sizes, the smaller pore volume leads to higher solution resistance, amplifying the IR drop and exacerbating the nonuniformity.

A more detailed discussion on this point has been provided in **Response to Comment 1**.

We have supplemented the manuscript with a more thorough mechanistic explanation and characterization experiments, and we have also revised this section accordingly.

Line 345 – 354 in the revised manuscript:

“Based on the well-established transmission line model, the nonuniform current distribution is primarily attributed to the solution resistance (R_s), which is influenced by both the pore size and electrolyte conductivity (Fig. 4d)^{47,48}. As the distance from the counter electrode increases, the cumulative R_s grows, leading to a larger solution potential drop ($\Delta\varphi_s$). Although the electrode potential (φ_m) remains constant throughout the depth, this difference in $\Delta\varphi_s$ results in variations in the local overpotentials (η), calculated as $\eta = \varphi_m - \Delta\varphi_s - \varphi_e$, where φ_e represents the equilibrium potential⁴⁹. This spatial variation in overpotential ultimately causes the observed nonuniform current distribution within the EMs. Considering the critical role of electrolyte concentration on R_s , its impact on the current distribution was quantitatively simulated (Supplementary Fig. 25).”

Other revisions and additions can be found in the **Response to Comment 1**.

7. P. 14, the authors discussed increasing electrolyte concentration as a way to illustrate the effects of pore size on current and activity. However, it is obvious that increasing electrolyte concentration decreases solution resistance, and therefore increases current in the cell. This does not seem to have anything to do with membrane pore size.

Thank you for your suggestion, and we apologize for any confusion caused. The experiments with different electrolytes were conducted **under the same current**, so variations in solution resistance would not influence the current in the cell. However, reduced electrolyte concentration increases the IR drop, which amplifies spatial gradients in electrolyte potential drops (ϕ_s) across the electrode. This results in pronounced nonuniform current distribution. We have supplemented an optical observation experiment to demonstrate the current distribution under different electrolyte concentrations while maintaining the same total current.

In the optical observation experiment (Supplementary Fig. 28), we employed bromothymol blue as a pH-sensitive indicator and used the same setup as the fluorescence observation experiment, where localized OH^- production correlates with current density. The reaction intensity across different regions of the Pt electrode was visualized under varying electrolyte concentrations at **a fixed current**. At high electrolyte concentration (300 mM), OH^- ions were uniformly generated across the entire Pt electrode surface (observed as dark regions), indicating even current distribution. In contrast, at low electrolyte concentration (2 Mm), OH^- production became highly localized in the region near the counter electrode, where excessive current density even led to observable H_2 bubble formation. These observations confirm that current density on highly conductive electrode surfaces is strongly influenced by $\Delta\phi_s$. In contrast, when a non-porous flat electrode is used (positioned directly opposite the counter electrode), the overpotential remains spatially uniform across the electrode surface, thus remaining unaffected by electrolyte concentration (Supplementary Fig. 29).

To improve clarity, we have revised the relevant statements in the revised main text and revised Supplementary Information.

Line 361 – 362 in the revised manuscript:

“This phenomenon can be attributed to the reduced R_s , which decreases the potential drop ($\Delta\phi_s$) across the pores.”

Line 369 – 374 in the revised manuscript:

“To further validate the influence of electrolyte concentration on current density distribution, an optical observation experiment was performed using a Pt electrode and a pH-sensitive probe to visualize the electrogenerated OH^- ions (Supplementary Fig. 28 and Movie 2). At a high electrolyte concentration of 0.3 M, OH^- ions were generated uniformly across the entire Pt electrode surface. In contrast, under low electrolyte

concentration conditions (0.002 M), OH^- production became highly localized in the region near the counter electrode.”

Line 378 – 385 in the revised manuscript:

“At a fixed current density (39.4 mA cm^{-2}), when the electrolyte concentration was increased from 0.02 M to 0.5 M, the pseudo-first-order kinetic constant increased from 1.8 min^{-1} to 4.2 min^{-1} (Fig. 4f). However, increasing the electrolyte concentration does not significantly affect the catalytic performance in the flow-by mode, where reactions are confined to the surface of the EM (Supplementary Fig. 29). This suggests that the main reason for the enhanced catalytic performance of EMs at high electrolyte concentrations is the change in R_s , which alters the current distribution within the pores.”

Page 34 in the revised Supplementary Information:

Supplementary Fig. 28. Optical observation of current density distribution on the Pt

electrode surface across electrolyte concentrations ranging from 2 mM to 300mM. (a) Photograph of electrode configuration. (b) Evolution of pH distribution at electrode interface. Bromothymol blue was employed as a pH probe and the same setup as in the fluorescence observation experiment was used. Current: 0.6 mA.”

8. Fig. 4h and related discussion: the authors present results of increasing the membrane thickness based on residence time. Since the residence time is calculated based on membrane thickness, the effect of the greater thickness is effectively canceled out, as the residence time has a normalizing effect. Therefore, this figure actually seems to show that the full length of the pores is utilized in the reaction. Otherwise, a constant residence time for different pore lengths would result in significant changes in activity, if the hypothesis of the authors is correct that only the area near the surface of the membrane is utilized for reactions. These results seem to disprove the proposed mechanisms by the authors, actually showing the opposite of the key scientific claim.

Thank you for your comment, and we sincerely apologize for any confusion caused. In the initial manuscript, Fig. 4h was plotted under the assumption that EMs with different thicknesses had identical volumes (which is not the real situation), aiming to provide directly comparable data. Consequently, at the same flow rate, EMs of different thicknesses exhibited the same HRT, as noted in the original figure caption.

To improve clarity, we have replotted the figure with flow rate as the x-axis. The revised results demonstrate that 1mm-EM₇ and 3mm-EM₇ achieve identical performance at the same flow rate, implying their equivalent effective depth. Additionally, we have included supplementary Fig. 30 in the Supplementary Information with HRT as the x-axis (calculated based on the actual electrode volumes). Notably, 1mm-EM₇ achieves catalytic efficiency equivalent to 3mm-EM₇ at 33% of the HRT (i.e., the same flow rate), further confirming the equivalence of their effective depth.

To improve clarity, we have revised the relevant statements in the main text and Supplementary Information.

Line 391 – 400 in the revised manuscript:

“It can be found that the thickness of EM₇ had little impact on its catalytic performance (Fig. 4h and Supplementary Fig. 30). Even when the thickness (and consequently the Ru loading) was reduced to one-third, the 1mm-EM₇ still exhibited an extraordinary

nitrate conversion rate. By comparing the C_{d1} values of EM₇ with different thicknesses (Supplementary Fig. 31), it is evident that the ECSA of 1mm-EM₇ is significantly lower than that of 3mm-EM₇. The discrepancy indicates that the ECSA cannot be fully utilized in microchannels with relatively high current densities and low solution conductivities. These findings suggest that EM₇ of varying thicknesses share a similar electroactive thickness, highlighting the potential to maintain high catalytic performance while reducing material usage and fabrication costs.”

Figure 4h in the revised manuscript:

Fig. 4 Current distribution in the channel of EMs. h Nitrate removal performance on EM₇ with different thicknesses.”

Page 36 in the revised Supplementary Information:

Supplementary Fig. 30. Nitrate removal performance on EM₇ with different thicknesses as a function of HRT.”

Page 37 in the revised Supplementary Information:

Supplementary Fig. 31. The cyclic voltammetry (CV) curves of EMs with different pore sizes and the corresponding electrical double-layer capacitance (C_{dl}). CV curves of (a) 2mm-EM₇, (b) 1mm-EM₇, and (c) C_{dl} of EM₇ with different thicknesses by plotting current variation against the scan rate to fit a linear regression. C_{dl} values were measured at the potential of 0.91 V vs. RHE. Electrolyte: 0.1 M Na₂SO₄.”

Page 42 in the revised Supplementary Information:

Supplementary Fig. 36. Simulated nitrate reduction performance on EM₇ at (a) different electrode thickness and (b) different concentrations of electrolyte. The inset image shows the corresponding pseudo-first-order kinetic constants.”

9. Lines 465-466, the HRT is determined based on the geometric volume. However, it should be determined by the pore volume, which corresponds to the void volume that is actually occupied by water flowing through the membrane. This will vary with pore size.

We appreciate your suggestion, as clarifying this issue is critically important for our study. In our research, all EMs were operated under identical current. If HRT were calculated using pore volume ($\text{HRT}^* = V_p/Q$, where V_p denotes pore volume and Q represents EM flux), smaller-pore EMs would exhibit lower flux at the same HRT^* . This would result in the electrolyte solution passing smaller-pore EMs receiving a higher charge per unit volume compared to larger-pore EMs at the same HRT^* , creating an inherently biased comparison. By calculating HRT based on total volume instead, we can avoid this bias. Therefore, for our system, using total volume provides a more appropriate basis for fair comparisons.

As you pointed out, variations in porosities among EMs indeed may affect catalytic performance due to the combined effects of mass transfer, reaction surface area and current distribution. When porosity is reduced while maintaining constant pore size, the specific surface area decreases, which increases local current density. This, in turn, amplifies the IR drop and exacerbates nonuniformity in current distribution. To investigate the effect of porosity, we partially blocked pores in EM₈₀ and EM₇ to reduce their porosity. When the porosity of EM₇ was reduced to 60% (matching that of EM₄), a 15% decline in catalytic performance was observed (Supplementary Fig. 38). Notably,

the catalytic efficiency of the modified EM₇ still outperformed EM₄ at the identical porosity, underscoring the critical contribution of spatial confinement effects.

Based on the above analysis, we have retained the original HRT determination method in the manuscript. However, we have expanded the Discussion section to include a detailed analysis of porosity's influence (Supplementary Fig. 38).

We have revised the relevant statements in the main text and Supplementary Information.

Line 491 – 496 in the revised manuscript:

“Porosity also plays a significant role in determining performance. When the porosity of EM₇ was reduced to 60% (matching that of EM₄) by selectively blocking a portion of its pores, a 15% decline in catalytic performance was observed (Supplementary Fig. 38), likely due to impaired current distribution and reduced reactive surface area. Nonetheless, EM₇ still outperformed EM₄ in reaction kinetics at the identical porosity, underscoring the critical contribution of spatial confinement effects.”

Page 44 in the revised Supplementary Information:

Supplementary Fig. 38. Catalytic performance of (a) EM₇ and (b) EM₈₀ with a 40% reduction in porosity. Current density: 39.3 mA cm⁻².”

10. What are the flow rates of the permeate through the membranes? Since the HRT is kept constant and the pore size is changing, presumably the flow rate for each is also changed. Therefore, the obtained volcano relationship could actually be due to

optimization of mass transfer, considering both diffusion and convection, as was previously shown for electrified membranes: doi.org/10.1038/s44221-024-00278-7

Thank you for your suggestions. Analyzing mass transfer in EMs is indeed essential. In the nitrate reduction experiments, the permeate flow rates through the membranes fall within the range of 0 – 0.09 mL cm⁻² s⁻¹. Since HRT is calculated based on the electrode volume, the flow rate for each EM remains identical under a fixed HRT. However, EMs with different pore sizes exhibit varying porosities, resulting in distinct flow velocities within their pores (while maintaining the same charge per unit solution during the reaction). The estimated maximum flow velocities in EMs are summarized in **Supplementary Table 5**.

Under these complex conditions, we employed a well-established method to evaluate the mass transfer capabilities of EMs with different pore sizes (*ACS Nano* **2019**, 13, 6998–7009 and *Electrochim. Acta* **1990**, 35, 1369-1376). Specifically, Cu²⁺ was used as a probe substance. By flowing Cu²⁺ solutions through EMs with varying pore sizes at controlled flow rates and a potential of 0.25 V_{RHE}, we quantified the mass transport rates via Cu²⁺ reduction currents. The results revealed a clear increase in mass transport rates with decreasing pore sizes (Supplementary Fig. 13). At low flow rates, Cu²⁺ was fully reduced across all EMs regardless of pore size, resulting in comparable mass transfer rate. However, at higher flow rates, larger-pore electrodes exhibited mass transport rates significantly below the convection limit (where all Cu²⁺ reaching the electrode undergoes complete reaction). Notably, EM₄ demonstrated higher mass transport rates than EM₇, indicating that the observed volcano-shaped relationship does not stem from mass transfer optimization.

Additionally, by transposing the Supplementary Fig. 13a, we can obtain a curve consistent with the schematic mass transfer curve in Fig. 5e. At a fixed flow rate, mass transfer efficiency gradually plateaus as pore size decreases (Supplementary Fig. 37). At this stage, the influence of current distribution becomes dominant, as described by the volcano-shaped curve. **In other words, although EM₄ exhibits the highest mass transfer rate, its catalytic performance is inferior to that of EM₇. Our results demonstrated that this contradictory is primarily attributed to the current nonuniformity caused by spatial confinement, as discussed above.**

We have revised the relevant statements in the main text and Supplementary Information.

Line 172 – 180 in the revised manuscript:

“To further elucidate the mass transport capabilities of the EMs, Cu^{2+} was employed as an electrochemical probe, as its reduction current effectively reflects mass transfer toward the catalyst surface^{38,39}. The results showed that the observed mass transport rate (R_{obs}) increased significantly as the pore size decreased (Supplementary Fig. 13). Notably, EM₄ exhibited the highest mass transport performance among all EMs, reaching the convection limit across the entire tested range. This paradox, where enhanced mass transfer coincides with diminished catalytic activity, implies that electron transfer processes may also play a critical role in determining the overall performance of the EMs.”

Line 477 – 480 in the revised manuscript:

“However, the enhancement in mass transfer becomes less prominent once the pore size drops below a certain threshold, as evidenced by both experimentally measured mass transfer rates (Supplementary Fig. 37) and simulated reactant concentration profiles (Fig. 5b).”

Line 545 – 556 in the revised manuscript:

“Mass transfer determination

The observed mass transfer rate (R_{obs}) value to the electrode was determined using a well-established approach^{38,39}. A solution containing 0.1 mM Cu^{2+} and 0.1 M Na_2SO_4 was used as the electrolyte. The same assembly as the flow-through experiments was employed to conduct all the tests. The counter and reference electrodes were a Pt wire and Ag/AgCl electrode, respectively. All the electrochemical experiments were performed on a CHI660E electrochemical workstation at 30 ± 1 °C. The reduction current of Cu^{2+} was obtained at 0.26 V_{RHE}. The R_{obs} value was calculated using equation (1)⁵⁰:

$$R_{\text{obs}} = \frac{I}{zFAC_{\text{b}}} \quad (1)$$

where I is the reduction current (A), z represents the number of electrons transferred (2 for the reduction of Cu^{2+}), F denotes the Faraday constant ($96,500 \text{ C mol}^{-1}$), A is the geometry surface area of the electrode ($2.54 \times 10^{-4} \text{ m}^2$), and C_{b} is the bulk concentration of Cu^{2+} (0.1 mol m^{-3}).”

Page 19 in the revised Supplementary Information:

Supplementary Fig. 13. Mass transfer rate on EMs. (a) Observed mass transfer rate (R_{obs}) on EMs. (b) Comparison of R_{obs} between EM₄ and EM₈₀. Highlighted areas correspond to the contribution of surface area and pore size. Since EM₄'s ECSA is 2.8 times that of EM₈₀, the orange region represents the maximum contribution (2.8×) from the specific surface area to mass transport, while the blue region reflects the minimal contribution from pore structure. Experiments were conducted at 0.26 V_{RHE} in an electrolyte containing 0.1mM Cu²⁺, and 0.1M Na₂SO₄.”

Page 43 in the revised Supplementary Information:

Supplementary Fig. 37. Mass transfer rate on EMs at different fluxes as a function of HRT.”

11. SI line 59, what concentration of nitrate is used? Is it the initial concentration, or the concentration of nitrate removed? For the energy consumption for conversion, it should be the latter. In the main text line 185, it seems like the nitrate conversion is assumed to be 80%, but the actual conversion should be included for calculating the energy consumption, not an assumed conversion.

We sincerely appreciate your suggestions. As you pointed out, the previous equation is incorrect and lacks clarity. When comparing the energy consumption of the EMs, we evaluated the total energy consumption (kWh g-N⁻¹) required to achieve an 80% single-pass nitrate removal efficiency, including both electrical energy consumption ($E_{\text{electrical}}$) and pumping energy (E_{pumping}). Specifically, we calculated the HRT needed to achieve this 80% removal rate for each EM using the rate constant (k_{obs}) (Supplementary equation (1)). Based on this HRT, we determined the volumetric flow rate (Q) employed in Supplementary equation (2). As you correctly noted, in Eq. 2 and Eq. 3, the nitrate concentration should be 80% of the initial concentration.

We have revised the relevant statements in the main text and Supplementary Information.

Line 212 – 218 in the revised manuscript:

“The energy consumption of the EMs required to achieve 80% nitrate removal efficiency was calculated (Fig. 2g and Supplementary Table 4). Notably, EM₇ and EM₈₀ required HRT of 35 s and 85 s, respectively, to reach this target. Under these conditions, the transmembrane pressure on EM₇ reached 6.0 kPa (Supplementary Fig. 18), which is 8.5 times higher than that of EM₈₀ (0.7 kPa). However, the contribution of pumping energy per unit of nitrate removed is negligible compared to electrical energy. For instance, the pumping energy for EM₇ was 0.003 kWh g-N⁻¹, accounting for only 0.7% of the total energy consumption (0.39 kWh g-N⁻¹).”

Page 3 in the revised Supplementary Information:

“The $E_{\text{electrical}}$ value was calculated according to Supplementary equation (2)¹,

$$E_{\text{electrical}} = 10^{-3} \times \frac{V_{\text{cell}} I}{QC} \quad (2)$$

where V_{cell} is the cell potential (V), I is the current used in the experiment (i.e., $2.54 \text{ cm}^2 \times 39.4 \text{ mA cm}^{-2} \times 10^{-3} = 0.1 \text{ A}$), Q is the volumetric flow rate at which 80% nitrate removal was achieved ($\text{m}^3 \text{ h}^{-1}$), C is the concentration of nitrate removed (80% of the initial concentration) (g-N m^{-3}).”

Page 49 in the revised Supplementary Information:

“Supplementary Table 4. Energy consumption for nitrate removal on EMs.^a

	Cell Voltage [V]	k [min ⁻¹]	HRT _{80%} ^b [min]	Flow Rate [10 ⁻² mL cm ⁻² s ⁻¹]	ΔP^c [kPa]	$E_{\text{electrical}}^d$ [kWh g-N ⁻¹]	E_{pumping}^e [kWh g-N ⁻¹]	Energy Consumption [kWh g-N ⁻¹]
EM ₄	3.4	2.10	0.75	0.46	9.8	0.4987	0.0045	0.5032
EM ₇	3.4	2.74	0.59	0.60	6.0	0.3895	0.0028	0.3923
EM ₂₄	3.5	1.86	0.87	0.40	3.0	0.5906	0.0014	0.5920
EM ₅₈	3.5	1.38	1.17	0.30	0.8	0.7961	0.0004	0.7964
EM ₈₀	3.5	1.14	1.41	0.25	0.7	0.9636	0.0003	0.9640

^aThe energy consumption was calculated according to the methods described in the Supplementary Text. The concentration of nitrate is 1 mM.

^bHRT_{80%} refers to the time required to reach 80% nitrate removal.

^c ΔP refers to the transmembrane pressure and is determined from Supplementary Fig. 18.

^d $E_{\text{electrical}}$ refers to the electrical energy. ^e E_{pumping} refers to pumping energy.”

12. It seems as though the reason for the FE for ammonia changing with pore size could be actually due to activity (FE for different products changes for different conversation rates). It would be more accurate to instead compare FE for nitrate conversion rather than for one product since the conversion changes.

We sincerely appreciate your suggestions. As you mentioned, the Faradaic efficiency (FE) for ammonia variation with pore size could be attributed to differences in the electrochemical activity of the EMs. In fact, due to the high selectivity (> 93%) of nitrate reduction to ammonia across all the EMs, the FE values calculated based on nitrate conversion are very close to those calculated based on ammonia production. Nevertheless, to improve accuracy, we have recalculated the FE for nitrate conversion to replace the previously reported FE for ammonia.

We have revised the relevant statements in the main text and Supplementary Information.

Line 202 – 211 in the revised manuscript:

“Based on the product analysis, the faradaic efficiencies (FE) for nitrate reduction were calculated. At an HRT of 3 s (Fig. 2f), the FE reached 28% on EM₇, indicating a higher electron utilization efficiency. In comparison, the FEs on EM₄ and EM₈₀ were 18% and

10%, respectively. Prolonging the HRT would reduce the FEs, mainly due to the overconsumption of nitrate and decreased flow rate. Nonetheless, the superiority of EM₇ remains evident, exhibiting a nearly doubled FE (15%) at 27 s HRT compared to EM₈₀ (8%). When the nitrate concentration was increased to 10 mM, all EMs showed enhanced FEs, with EM₇ reaching a peak value of 80% under a 7 s HRT (Supplementary Fig. 17). As mass transfer limitations are alleviated at higher concentrations, the discrepancies in FEs among EMs become less pronounced.”

Figure 2f in the revised manuscript:

Fig. 2 Nitrate reduction performance on EMs. f FEs for nitrate reduction on the EMs at HRTs of 3 and 27 s. Nitrate concentration: 1 mM.”

Page 23 in the revised Supplementary Information:

Supplementary Fig. 17. Faradaic efficiencies (FE) for nitrate reduction on EMs at HRTs of 3, and 27 s. Nitrate concentration: 10 mM. Electrolyte: 0.1 M Na₂SO₄ (pH = 7).”

13. The authors find an optimal pore size for their system, but it is not clear how this finding could be easily extrapolated to any other systems, as there are no quantitative relationships or fundamental/generalizable parameters drawn from their study that seemed like it could be easily extended to other systems.

We sincerely appreciate your valuable suggestions, which are crucial for improving the scientific merit of our work. In this study, we investigated the key factors influencing EMs through experimental and finite element analysis (FEA) methods. Although intriguing trends have been derived, it is still challenging to establish straightforward quantitative relationships. Nevertheless, to gain deeper insights into the quantitative mechanism, we have systematically summarized the impact of critical design and operational parameters (e.g., pore size, applied current, catalytic activity, electrode porosity, diffusion coefficient and spacing between electrodes) on the catalytic performance of EM reactors (Supplementary Fig. 38 and 39) through combined modeling and experimental approaches.

Our analysis reveals that pore size, current density and electrolyte concentration play important roles in the catalytic performance of EMs. The findings about these generalizable parameters provide a framework for the rational design of EM reaction systems in future studies.

We incorporated the relevant statements into the discussion section.

Line 486 – 505 in the revised manuscript:

“Through combined simulations and experiments, we systematically analyzed the influence of EM architecture and electrolyte properties on catalytic performance. Although thickness directly impacts the surface area and catalytic loading, the benefit of increasing thickness is marginal, as it does not necessarily enhance reactive depth. Additionally, a high electrolyte concentration is essential for promoting current uniformity and expanding the effective reaction area. Porosity also plays a significant role in determining performance. When the porosity of EM₇ was reduced to 60% (matching that of EM₄) by selectively blocking a portion of its pores, a 15% decline in catalytic performance was observed (Supplementary Fig. 38), likely due to impaired current distribution and reduced reactive surface area. Nonetheless, EM₇ still outperformed EM₄ in reaction kinetics at the identical porosity, underscoring the critical contribution of spatial confinement effects. The impact of other key design parameters, including applied current, intrinsic catalytic activity, reactant diffusion coefficient, and electrode spacing, were also analyzed by simulation, respectively (Supplementary Fig. 39). As the applied current increases, faster mass transfer is required to balance electron

transfer, which amplifies the advantage of smaller pores and may reduce the optimal pore size. Meanwhile, improvements in intrinsic catalytic activity enhance surface reaction rates but have minimal influence on the optimal pore size. Similarly, variations in diffusion coefficient and electrode spacing show little effect on the optimal pore dimension. This in-depth understanding of the structure-performance relationship offers new insights into the design of EMs with ultrahigh reaction kinetics, advancing beyond conventional catalyst development paradigms.”

Page 44 in the revised Supplementary Information:

Supplementary Fig. 38. Catalytic performance of (a) EM₇ and (b) EM₈₀ with a 40% reduction in porosity. Current density: 39.3 mA cm⁻².”

Page 45 in the revised Supplementary Information:

“

Supplementary Fig. 39. Simulated reaction kinetics of Sim4, Sim7, and Sim24 under varied conditions: **(a)** applied current, **(b)** reactant diffusion coefficient, **(c)** hydrogen evolution reaction exchange current density, **(d)** nitrate reduction exchange current density, and **(e)** electrode spacing. The y-axis represents the ratio of k_{obs} of sim4 or sim24 to the k_{obs} of Sim7 under an identical condition. $D = 1 \times 10^{-9} \text{ m}^2 \text{ s}^{-1}$.

MINOR COMMENTS

1. Some references do not seem to be appropriate for the contents, such as references 23, 24 and 25 in line 57.

Thank you for your careful review. We have re-evaluated the references and revised some, such as references 11, 20, 23, 24, and 25, to better align with the content.

2. In figure S 13, which product is the current efficiency referring to?

We apologize for any confusion caused. Here, we refer to the FE for TCAA conversion. To improve clarity, we have revised the corresponding description in the Supplementary Information.

Page 26 in the revised Supplementary Information:

Supplementary Fig. 20. FEs for TCAA conversion on EMs at HRTs of 3, and 27 s. TCAA concentration: 1 mM. Electrolyte: 0.1 M Na₂SO₄ (pH = 7).”

Point-by-point response to the reviewers' comments

Title: “*Confinement-enhanced valorization of contaminants in electrified hydrogenation membranes for water purification*”

Manuscript ID: NCOMMS-24-81216A

We extend our sincere gratitude to the reviewers for their insightful comments and constructive suggestions, which have significantly contributed to enhancing the quality of this work. We have carefully and systematically responded to all the points raised. The reviewers' comments are in ***bold italic font*** and our revisions are in blue font. We have also highlighted the revised text in blue in the main text. Below are our detailed responses to each point provided.

REVIEWER COMMENTS

Reviewer #1 (Remarks to the Author):

This reviewer is satisfied with the response, and has no further comment.

Many thanks to the reviewer for the recognition of this work. Your important review greatly helps to improve the clarity and quality of our manuscript.

Reviewer #2 (Remarks to the Author):

The authors have done a good job of addressing most of the comments, specifically through running additional experiments and providing more details and explanations. Crucially, they conducted ECSA measurements, referencing electrode potentials to a reference electrode, and improved consistency for parameters used to calculate energy consumption. More importantly, I believe that they have now provided good evidence that the effect of pore size on current distribution through the electrified membranes influences reaction performance, which is the key finding and novelty in this work. I have some remaining comments, below, but I believe that if they are addressed (especially comment 1) then the manuscript would be suitable for publication in Nature Comms.

We sincerely appreciate your recognition and additional comments. Your thorough review and insightful suggestions will significantly improve the quality of our work. Below, we provide point-by-point responses to all the issues raised.

1. Reviewer 1, comment 4: The authors conclude that since mass transfer improves with decreasing pore size, and since EM4 with the smallest pore size shows reduced reaction activity, then other factors such as nonuniformity in current could explain the trends. However, the results show that EM4 reaches the convection limit. Therefore, could it be that EM4 is simply additionally mass transport limited by convection, leading to the lower reaction activity? Further, it appears that mass transport must account substantially for the reported decrease in activity with increasing pore size that occurs for EM7 – EM80. The authors should address how much this trend is due to changes in mass transport vs. current distribution (for example, by modifying parameters to ensure the same mass transport rate through all EMs; the same HRT in membranes that have different mass transport rates does not account for this).

We sincerely appreciate your valuable comments. As you noted, clearly distinguishing the respective contributions of mass transport and current distribution would significantly enhance the rigor of this study. Below, we provide detailed responses to your comment.

Regarding your first concern, we apologize for any lack of clarity. While EM₄ reaches the convection limit, this does not imply its reactivity is limited by convection in the reaction conditions. Specifically, the term “convection limit” here refers to the mass transport rate at which Cu²⁺ ions are completely consumed. Therefore, it represents the maximum achievable mass transfer rate (R_{obs}) under a given flow rate, a definition adapted from *J. Appl. Electrochem.* **1998**, 28, 697–702 and *ACS Nano* **2019**, 13, 6998–7009. This indicates that EM₄ exhibits superior reactant accessibility at the

electrode interface compared to other EMs at the same flux. Consequently, mass transport does not impose limitations on the reaction kinetic for EM₄.

In addition, to isolate the effect of current distribution on the reaction kinetics of EMs, we conducted batch experiments (recirculating permeated solution to the reservoir for continuous feeding) under controlled mass-transport conditions (Supplementary Fig. 40). By adjusting the flow rates across EMs with different pore sizes, an identical mass transfer rate can be achieved for all EMs. Results demonstrate progressively enhanced reaction kinetics with increasing pore size, where EM₈₀ outperformed EM₇. This performance disparity highlights the negative influence of nonuniform current distribution on small-pore EMs under same mass transport conditions.

We further investigated the influence of mass transport on EMs' reaction kinetics under a homogenized current distribution. Given that electrochemical experiments inevitably lead to nonuniform current distributions, we quantified this effect through simulations (Supplementary Fig. 39). The result revealed an enhancement in reaction kinetics as the pore size decreased, with EM₄ maintaining its superiority over the other EMs, confirming its highest mass transfer rate.

Quantitatively, the kinetic constant increased by 91% as the pore size decreased from 80 to 7 μm due to mass transport enhancement (Supplementary Fig. 39). By contrast, Supplementary Fig. 40 shows that EM₇ underperformed EM₈₀ by 16% due to current distribution nonuniformity. In this case, the effect of mass transport is evidently more significant than that of current distribution. When comparing EM₄ and EM₇, EM₄ demonstrates only a marginal 1% advantage in terms of mass transfer (Supplementary Fig. 39), while it is 19% inferior to the latter due to more pronounced current distribution nonuniformity (Supplementary Fig. 40). Here, the influence of current distribution dominates over that of mass transport. The interaction of these two factors ultimately accounts for the superior performance of EM₇ relative to other EMs. Overall, these comparisons allow for a semi-quantitative distinction of the contributions from mass transfer and current uniformity. It is also noteworthy that the differences in mass transfer among the EMs were underestimated in this comparison, as the mass transfer region was also expanded due to the uniform current distribution.

We have added the relevant discussion in the main text and Supplementary Information.

Line 177 – 179 in the revised manuscript:

“Notably, EM₄ exhibited the highest mass transport performance among all EMs, reaching the convection limit (i.e., Cu²⁺ is fully consumed at a specific flux) across the entire tested range.”

Line 482 – 490 in the revised manuscript:

“However, the enhancement in mass transfer becomes less prominent once the pore size drops below a certain threshold, as evidenced by both experimentally measured mass transfer rates (Supplementary Fig. 38) and simulated reaction kinetics at uniform current distribution (Supplementary Fig. 39). Meanwhile, the nonuniformity of current distribution along the channel depth was intensified as the pore size decreased, due to the increasing solution resistance inside the channel. As demonstrated by the batch mode experiments conducted under equivalent mass transfer conditions (Supplementary Fig. 40), such nonuniformity significantly reduces the catalytic performance of EMs with smaller pore sizes.”

Page 45 in the revised Supplementary Information:

Supplementary Fig. 39. The simulation of EM channels with uniform current distribution. (a) The simulated nitrate removal performance when all model channels have a uniform current distribution. (b) The corresponding pseudo-first-order kinetic constants of nitrate reduction on Sims. To exclude any possible impact of nonuniform current distribution, we employed a first-order reaction model to represent nitrate reduction on the pore surface. The kinetic constant of the surface reaction was set to be proportional to the average local current density and kept uniform throughout the simulated channel.”

Page 46 in the revised Supplementary Information:

Supplementary Fig. 40. The performance of EMs with the same mass transfer rate at a batch mode. (a) The nitrate removal performance on EMs with the same mass transfer rate at a batch mode. (b) The corresponding pseudo-first-order kinetic constants of EMs. The experiments were carried out at batch mode (recirculating permeated water to the reservoir for continuous feeding) with a total liquid volume of 20 mL. The flux of each EM was adjusted to ensure they have the same mass transfer rate (0.07 cm s^{-1}). The flux of EM₈₀, EM₅₈, EM₂₄, EM₇, and EM₄ was 0.30, 0.10, 0.08, 0.07, and $0.07 \text{ mL cm}^{-2} \text{ s}^{-1}$, respectively, according to Supplementary Fig.13.”

2. The authors noted that they reduced TiO₂ in hydrogen at elevated temperature and that the sample becomes conductive. Therefore, I believe that the confusion arises because this is a titanium suboxide: for example, many EM papers generate Ti_{4O₇} as a highly conductive Magneli phase titanium oxide material. It is therefore now inaccurate to refer to the synthesized material (after the hydrogen treatment) as TiO₂. They could refer to it instead as reduced titania/titanium oxide or titanium suboxide.

We appreciate your insightful observation. Although numerous oxygen vacancies were introduced into TiO₂ after the thermal annealing treatment in a hydrogen atmosphere, the XRD data shown in Fig. 1h indicates that the material maintained most of its original crystal structure (i.e., neither Ti_{4O₇} nor Magneli phase TiO₂). To ensure both terminological accuracy and conciseness, we have adopted TiO_{2-x} throughout the revised manuscript, aligning with a previously published article in the field of electrocatalytic materials (*Nat. Commun.* **2025**, 16, 1122). The changes have been highlighted in blue font in the revised manuscript.

3. The authors have done a great job of adding more explanation and experimental verification of the variation in current through the depth of the EMs, which is central

to the key findings and novelty. As a follow-up to this: the authors should consider that the changes in limiting diffusion current through the EMs likely varies with EM pore size (presumably the reactions are faster in the smaller pores, so the limiting diffusion current will likely increase as well), which should affect the current distribution. In other words, pore size influences mass transport, so the influence of mass transport on current density should also change with pore size.

We appreciate your insightful comment. As you pointed out, since pore sizes influence both mass transfer and current distribution, the interaction between these two factors will change accordingly with pore size. In our experiment, all EMs were tested under a **fixed total current**. There are two parallel reactions occurring on pore surfaces:

(1) The hydrogen evolution reaction, which is considered unaffected by mass transport;

(2) The nitrate reduction reaction, which is influenced by mass transfer.

The total applied current, maintained consistently across all EMs, equals the summation of currents from these two parallel reactions.

Direct experimental observation of pore-scale diffusion effects on nitrate reduction current distribution remains challenging. Nevertheless, the nitrate reduction current distribution profiles can be separately derived from the simulation results, which allows us to explicitly investigate the effect of mass transport on current distribution across varying pore sizes, as shown in the updated Supplementary Fig. 25.

The simulations reveal that the current of nitrate reduction is concentrated near the electrode surface. This nonuniform and localized distribution of current, combined with the decreasing nitrate concentration at the reaction interface, results in a peak in the nitrate reduction current, which is indicative of a diffusion-limited process. As the pore size decreases, the enhanced mass transfer rate and intensified current nonuniformity lead to a shift of the current peak position closer to the surface. It is important to note that, although small-pore EM exhibits a higher overall limiting diffusion current due to their faster mass transport rate, their local peak current is relatively lower, which is primarily attributed to their larger specific surface area.

We have added the relevant statements/discussion in the main text and Supplementary Information.

Line 330 – 333 in the revised manuscript:

As pore size decreases, nitrate reduction current also becomes more nonuniform (Supplementary Fig. 25), which, together with the decreasing nitrate concentration along the channel, leads to the formation of a current peak. Reduction in pore size shifts this peak toward the near end due to enhanced mass transfer and increased current nonuniformity.”

“**Supplementary Fig. 25.** The simulated nitrate reduction current distribution at an HRT_{geom} of 91 s. The inset image presents an enlarged view of the current distribution profile, illustrating that the peak current shifts closer to the surface as the pore size decreases.”

4. Reviewer 2, comment 9: *The authors explain that calculating HRT based on pore volume rather than geometric surface area would result in an unfair comparison, as smaller pores results in lower flux at the same HRT (although I believe they meant that it results in higher flux). Instead of using geometric volume, which does not in fact represent the volume of water passing through the membrane, perhaps the comparisons should be based on flow rate, which should be more consistent among the membranes with different pore volumes. The authors also do not seem to explain why geometric volume is a valid way to calculate HRT, other than that it's an alternative that avoids concerns with using pore volume. Specifically, what is the physical meaning of HRT based on geometric volume in these porous systems?*

Thank you for your comment. We apologize for any ambiguity in our previous manuscript and response that may have caused confusion and misunderstanding. To improve clarity, we would like to explicitly clarify that the term “flux” used both in the manuscript and response letter refers specifically to **the volumetric flow rate of liquid passing through a unit membrane area per unit time ($mL\ cm^{-2}\ s^{-1}$)**. This definition follows the standard terminology in membrane-based water treatment and other industrial fields. It should be noted that **“flux” in this context does not represent the velocity of liquid within the pores.**

In addition, two definitions of HRT may be employed, one based on the geometric volume (here denoted as HRT_{geom}) and the other based on the pore volume (here denoted as HRT_{pore}). The relationship between flux and HRT can be expressed by the following equation:

$$\text{Flux} = \frac{\text{geometric volume}}{(\text{surface area}) \times \text{HRT}_{\text{geom}}} = \frac{\text{pore volume}}{(\text{surface area}) \times \text{HRT}_{\text{pore}}}$$

In terms of physical meaning, while HRT_{pore} corresponds to the residence time within the pore, HRT_{geom} represents the nominal reaction time from a reactor perspective. Both the two terms are widely recognized and have been adopted in previous studies. For instance, HRT_{geom} has been employed in *Water Res.* **2022**, 224, 119047 and *Adv. Mater.* **2023**, 2310954 and serves as a standardized metric that allows for comparison among different membranes. Although HRT_{pore} has been also employed in numerous studies (*Proc. Natl. Acad. Sci. U. S. A.* **2023**, 120(11), e2217703120 and *Nat. Nanotechnol.* **2023**, 18, 160–167), these studies typically do not involve comparative analyses of electrocatalytic performance across membranes with different pore sizes.

In our experiments with EMs of varying pore sizes, an identical current (100 mA) was applied. If HRT_{pore} was employed, the flux would vary for different EMs at the same HRT_{pore} due to the variation of pore volume. This would lead to a mismatch between the transferred charge and the treated solution volume, making it impossible to compare reaction kinetics accurately.

By using HRT_{geom} as an indicator to calculate reaction kinetics, the relationship between transferred charge and reacted solution volume can be normalized among EMs. Besides, this term also enables a meaningful comparison of reaction kinetics across different EMs and facilitates future reactor design. Although employing HRT_{geom} may result in “unfair” comparison in specific parameters (e.g., liquid velocity inside the pores), it imposes little impact on the core principles underlying this study. Therefore, the use of HRT_{geom} is an appropriate approach for our research.

As you pointed out, flow rate is indeed a good parameter to consider. We have incorporated this parameter in figures related to mass transfer, such as Supplementary Fig. 13. When analyzing the reaction kinetics, HRT_{geom} was adopted as a time-based metric. The use of these parameters is consistent with other studies in the field of electrified membranes.

To enhance clarity and rigorousness, we have revised the expression of HRT in the manuscript. The nominal HRT calculated from the geometric volume is now denoted as “ HRT_{geom} ”. We have also added the corresponding explanations in the main text.

Line 551 – 552 in the revised manuscript:

“The HRT_{geom} was determined based on the geometric volume of the EMs (0.76 cm³) in order to correct for porosity variations among the EMs.”

An additional discussion regarding the influence of porosity was included in the previous revision.

Line 499 – 504 in the revised manuscript:

“Porosity also plays a significant role in determining performance. When the porosity of EM₇ was reduced to 60% (matching that of EM₄) by selectively blocking a portion of its pores, a 15% decline in catalytic performance was observed (Supplementary Fig. 41), likely due to impaired current distribution and reduced reactive surface area. Nonetheless, EM₇ still outperformed EM₄ in reaction kinetics at the identical porosity, underscoring the critical contribution of spatial confinement effects.”

Page 47 in the revised Supplementary Information:

Supplementary Fig. 41. Catalytic performance of (a) EM₇ and (b) EM₈₀ with a 40% reduction in porosity. Current density: 39.3 mA cm⁻².”

5. Reviewer 2 comment 11: *If HRT is used to determine volumetric flow rate, but the volume used to calculate HRT is the geometric volume and not the pore volume, then the obtained flow rate should be inaccurate, resulting in inaccurate calculation of the energy efficiency.*

Thank you for your valuable comment. We apologize for any misunderstanding caused. The volumetric flow rate (Q) refers to the **total volume of liquid (m³ h⁻¹) passing through the EMs** and can be calculated as:

$$Q = \frac{\text{pore volume}}{\text{HRT}_{\text{pore}}} = \frac{\text{geometric volume}}{\text{HRT}_{\text{geom}}}$$

The volumetric flow rate was determined based on both the geometric volume and the corresponding HRT_{geom}, ensuring the accuracy of the calculated volumetric flow rate and the corresponding energy consumption. Specifically, the HRT_{geom} values required to achieve an 80% removal rate for EM₄, EM₇, EM₂₄, EM₅₈, and EM₈₀ were 45, 35, 53, 70, and 85 s, respectively. The corresponding volumetric flow rates were

0.61, 0.78, 0.53, 0.39, and $0.32 \times 10^{-4} \text{ m}^3 \text{ h}^{-1}$, respectively. Finally, the electrical energy consumption associated with achieving 80% nitrate removal can be calculated based on these values.

We have standardized the terminology throughout the manuscript to ensure clarity. The concept of volumetric flow rate is exclusively used in the context of energy consumption calculations. The manuscript does not involve the concepts of velocity or flow rate within the pores, except where necessary for estimating the Reynolds number. The equation for calculating the volumetric flow rate has also been provided in the Supplementary Information.

Page 3 in the revised Supplementary Information:

“The Q value was calculated by Supplementary equation (3),

$$Q = 3600 \times \frac{V_{\text{geom}}}{\text{HRT}_{\text{geom},80\%}} \quad (3)$$

where V_{geom} is the geometric volume of the electrified membrane ($7.6 \times 10^{-7} \text{ m}^3$).”

The calculated volumetric flow rate has been added to Supplementary Table 4:

“Supplementary Table 4. Energy consumption for nitrate removal on EMs.^a

	Cell Voltage [V]	k_{obs} [min ⁻¹]	$\text{HRT}_{\text{geom},80\%}^{\text{b}}$ [s]	Q [$10^{-4} \text{ m}^3 \text{ h}^{-1}$]	ΔP^{c} [kPa]	$E_{\text{electrical}}^{\text{d}}$ [kWh g-N ⁻¹]	$E_{\text{pumping}}^{\text{e}}$ [kWh g-N ⁻¹]	Energy Consumption [kWh g-N ⁻¹]
EM ₄	3.4	2.10	45	0.61	9.8	0.4987	0.0045	0.5032
EM ₇	3.4	2.74	35	0.78	6.0	0.3895	0.0028	0.3923
EM ₂₄	3.5	1.86	52	0.53	3.0	0.5906	0.0014	0.5920
EM ₅₈	3.5	1.38	70	0.39	0.8	0.7961	0.0004	0.7964
EM ₈₀	3.5	1.14	85	0.32	0.7	0.9636	0.0003	0.9640

^aThe energy consumption was calculated according to the methods described in the Supplementary Text. The concentration of nitrate is 1 mM.

^b $\text{HRT}_{\text{geom},80\%}$ refers to the time required to reach 80% nitrate removal.

^c ΔP refers to the transmembrane pressure and is determined from Supplementary Fig. 18.

^d $E_{\text{electrical}}$ refers to the electrical energy.

^e E_{pumping} refers to pumping energy.”

Point-by-point response to the reviewers' comments

Title: “*Confinement-enhanced valorization of contaminants in electrified hydrogenation membranes for water purification*”

Manuscript ID: NCOMMS-24-81216B

We extend our sincere gratitude to the reviewer for the insightful comment and constructive suggestion, which have significantly contributed to enhancing the quality of this work. We have carefully and systematically responded to the point raised. The reviewer's comment is in ***bold italic font*** and our revisions are in **blue** font. We have also highlighted the revised text in **blue** in the main text. Below is our detailed response to the point provided.

REVIEWER COMMENTS

Reviewer #2 (Remarks to the Author):

The revised manuscript addresses the points raised well. The additional experiments and simulations that separately evaluate the influences of mass transfer and current distribution now provide convincing evidence for the key result of optimal pore size that balances these effects. However, these tradeoffs are still not quantitatively evaluated in this manuscript, and it seems like mass transport is still the dominant aspect determining the pore size effect on conversion of reactants in solution. Therefore, I believe that the manuscript would be strengthened by adding some brief discussion about future work being required to quantitatively evaluate the respective contributions of these two phenomena in EMs. For example, there may be a more complex relationship among pore size effects, mass transport, and current distribution (which are all interrelated), and it may be that reducing pore size into the nm scale leads to different effects. Overall, though, this manuscript does provide convincing evidence for the concept of current nonuniformity being a limiting factor for smaller pore sizes leading to enhanced activity, which is a novel and valuable contribution for this field.

We sincerely appreciate your recognition and comment. Your thorough review and insightful suggestions will significantly improve the quality of our work.

As you noted in the comment, our revised manuscript separately evaluates the influences of mass transfer and current distribution. While the effect of mass transfer is more dominant at larger pore size (i.e., 7–80 μm), current distribution becomes the

decisive limitation for pores smaller than 7 μm . Although we have made considerable efforts to quantitatively investigate the trade-off between current distribution and mass transfer, fully decoupling these two interrelated processes remains challenging. To strengthen this aspect, we have emphasized the separate effects of current distribution versus mass transport and incorporated perspectives for future work in the Discussion section, outlining pathways to distinguish their respective contributions both mathematically and experimentally. Additionally, we discuss potential nanoconfinement effects in nanopores (<100 nm), where phenomena such as electric double layer overlap may reshape the relationship between pore size and electrocatalytic activity. The changes are listed below.

Line 390–392 in the revised Manuscript:

“Therefore, in addition to mass transport, current distribution emerges as a critical factor governing the catalytic activity of EMs. Its optimization represents a promising strategy for enhancing reaction kinetics.”

Line 449–450 in the revised Manuscript:

“Therefore, current distribution plays a more dominant role within this pore size range.”

Line 480–497 in the revised Manuscript:

“This study presents the design principle of EMs for efficient wastewater remediation and pollutant valorization by gaining mechanistic insights into the spatial confinement effect. EMs loaded with atomically dispersed Ru sites were fabricated, showcasing versatile activity for the hydrogenation of nitrate, TCAA, and phenol. Reducing the pore diameter of EM (e.g., EM₇ vs. EM₈₀) significantly decreases the diffusion lengths of reactants, leading to accelerated mass transfer and alleviated concentration polarization. However, the enhancement in mass transfer becomes less prominent once the pore size drops below a certain threshold (i.e., $\sim 7 \mu\text{m}$), as evidenced by both experimentally measured mass transfer rates (Supplementary Fig. 38) and simulated reaction kinetics at uniform current distribution (Supplementary Fig. 39). Meanwhile, the nonuniformity of current distribution along the channel depth was intensified as the pore size decreased, due to the increasing solution resistance inside the channel. As demonstrated by the batch mode experiments conducted under equivalent mass transfer conditions (Supplementary Fig. 40), such nonuniformity significantly reduces the catalytic performance of EMs with smaller pore sizes (e.g., EM₄ vs. EM₇). As a result of the compromise between mass transfer (favored in small-pore EM) and electron transfer (favored in large-pore EM), a volcano-shaped

relationship between electrochemical activity and membrane pore size was observed (Fig. 5e). Consequently, there exists an optimal pore size for EMs, which is determined to be around 7 μm in our system.”

Line 518–545 in the revised Manuscript:

“Collectively, our study reveals that while enhanced mass transport dominates the activity improvement at larger pore scales (e.g., EM₇ vs. EM₈₀), severe current nonuniformity becomes the primary limitation for smaller pores (e.g., EM₄). The electrochemical model provides a foundational framework that integrates current distribution, mass transfer, and electron transfer processes, enabling quantitative prediction of EM performance trends under varied conditions. For instance, the performance of novel EMs (e.g., asymmetric EM) can be explored *in silico* prior to experimental investigation. Given the superior catalytic efficiency of EMs, this study presents an instructive approach for designing cost-effective EMs. These advancements have significant implications not only for wastewater treatment but also for resource recovery, fostering the development of more sustainable wastewater management frameworks.

To further refine and quantify the insights provided by this framework, future work should aim to decouple the quantitative contributions of mass transport and current distribution. From a computational perspective, more intuitive mathematical models could be developed to directly account for the interplay between pore structure, mass transfer, and current distribution, thereby facilitating the rational design of optimal pore architecture. To enhance the model’s predictive capability, it is preferable to include the effects of bubbles, intrinsic reaction kinetics, and other relevant factors. Experimentally, an important research focus lies in the design of EM systems capable of isolating the individual influences of these interrelated processes. One promising approach is the fabrication of EMs with precisely tunable pore characteristics (e.g., size and porosity), using techniques such as laser etching or photolithography. On these well-defined platforms, tailoring the conductivity of electrolytes and electrode materials, combined with experiments performed under carefully controlled conditions (e.g., specific current densities and potentials), can help evaluate the influence of current distribution. Concurrently, *in situ* analytical methods, such as microfluidic chips, can be employed to quantitatively investigate the mass transfer at the electrode interface. Finally, extending studies to nanoscale pores (<100 nm) may also reveal distinct nanoconfinement effects (e.g., electric double layer overlap⁵⁰ and mesoscopic mass transfer¹⁵), which significantly influence both mass transport and local reaction kinetics, potentially reshaping pore-size optimization strategies.”

Response to the reviewers' comments

Title: *“Confinement-enhanced valorization of contaminants in electrified hydrogenation membranes for water purification”*

Manuscript ID: NCOMMS-24-81216C

REVIEWERS' COMMENTS

We sincerely thank the reviewers for their time and effort in reviewing our work. We greatly appreciate their insightful comments and constructive suggestions, which have significantly enhanced the quality of this work and encouraged us to further advance our research.